# Resting Neurons, Active Insights:
# Robustifying Activation Sparsity in LLMs via Spontaneity

**Haotian Xu** [1]   **Jiannan Yang** [1]   **Tian Gao** [2]   **Tsui-Wei Weng** [3]   **Tengfei Ma** [1]

## Abstract

Activation sparsity offers a compelling route to accelerate large language model (LLM) inference by selectively suppressing hidden activations, yet existing approaches exhibit severe accuracy degradation at high sparsity. We show that this failure stems from representational instability: *activation sparsity disrupts input-dependent activation learned during pretraining, inducing distribution shifts in hidden states.* We address this issue by reframing activation sparsity as a representational alignment problem and introducing **Spontaneous Neurons (SPON)**, a lightweight mechanism inspired by spontaneous neural activity in biological systems. SPON injects a small set of learnable, input-independent activation vectors that act as persistent representational anchors for sparse computation. These vectors are trained via distribution matching to the dense model and can be absorbed into bias terms after training, incurring negligible inference overhead. Across multiple LLM backbones, SPON consistently restores performance, stabilizes latent representations, and preserves generalization. Our results establish SPON as an effective and principled solution for reliable activation-sparse inference, and offer new insights into knowledge retention in LLMs.

## 1. Introduction

Large Language Models (LLMs) have achieved remarkable performance across a wide range of tasks (Achiam et al., 2023; Anthropic, 2024; Guo et al., 2025; Bai et al., 2023; Dubey et al., 2024), largely driven by scaling model

[1]Stony Brook University, Stony Brook, USA [2]IBM Thomas J. Watson Research Center, Yorktown Heights, USA [3]Halıcıoğlu Data Science Institute, UC San Diego, La Jolla, USA. Correspondence to: Tengfei Ma <tengfei.ma@stonybrook.edu>, Haotian Xu <haotian.xu@stonybrook.edu>.

*Proceedings of the 43rd International Conference on Machine Learning*, Seoul, South Korea. PMLR 306, 2026. Copyright 2026 by the author(s).

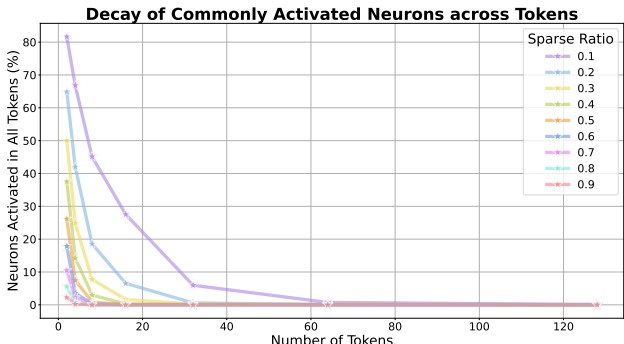

*Figure 1.* Fraction of neurons activated across all tokens decreases with sequence length under different activation sparsity levels.

capacity and training data (Brown et al., 2020). Despite these gains, the growing size of modern LLMs poses significant challenges for efficient inference and mechanistic interpretability, particularly in latency-sensitive and resource-constrained settings. To address these issues, a broad class of pruning and sparsity techniques has been developed (Sun et al., 2023; Frantar & Alistarh, 2023; Sawmya et al., 2024; Liu et al., 2024; Wang et al., 2024; Yin et al., 2023; Ma et al., 2023). Among them, activation sparsity methods are especially appealing, as they reduce runtime computation by selectively suppressing hidden activations (Liu et al., 2023; Wang et al., 2024; Liu et al., 2024).

However, achieving high sparsity levels often entails significant accuracy degradation or necessitates costly retraining and architectural modifications (Mirzadeh et al., 2023; Zhang et al., 2024b). While recent methods, such as TEAL (Liu et al., 2024), LaRoSA (Liu et al., 2025a), and R-Sparse (Zhang et al., 2025), optimize the efficiency-accuracy trade-off via magnitude thresholding and structured patterns, sparse models still consistently underperform their dense counterparts, indicating that selective inhibitions can disrupt pretrained representations and introduce instability or information loss. We hypothesize that this instability stems from the absence of universal activations that anchor pretrained representations during sparse computation. This hypothesis finds supported in the observation that, as shown in Figure 1, the proportion of neurons that remain permanently active across tokens decays exponentially with sequence length

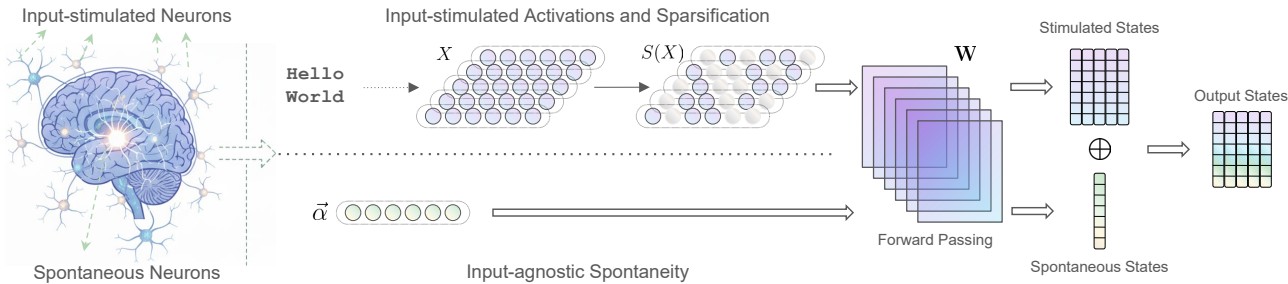

*Figure 2.* **Overview of Input Sparsification and Spontaneous Neurons.** Input activation sparsity masks low-magnitude entries to zero, acting analogously to column-wise pruning in weight matrices and avoiding register-level weight movement, thereby enabling wall-clock speed-ups. Complementarily, spontaneous neurons are injected via input-agnostic activations within each block.

under different sparsity levels (see Appendix A).

To address this limitation, we introduce **spontaneous neurons**, a lightweight mechanism inspired by spontaneous activity in biological neural systems (Hubel & Wiesel, 1962; Arieli et al., 1996; Kenet et al., 2003), to augment activation-sparse LLMs with few *input-independent activations* that serve as stable representational anchors. These spontaneous neurons complement input-dependent activations by encoding prior expectations distilled from the dense model, thereby mitigating the representational drift induced by sparsification. Importantly, they can be absorbed into bias terms after training, incurring no additional inference cost.

By reframing activation sparsity through the lens of representational stability, our work provides a principled approach to improving sparse inference in LLMs. Our contributions are threefold:

- (i) proposing spontaneous neurons as a biologically inspired and computationally negligible mechanism to stabilize activation-sparse LLMs;

- (ii) narrowing the performance gap between sparse and dense models across architectures, benchmarks, and sparsity levels at 1.178% improvement on average without sacrificing efficiency;

- (iii) providing rigorous theoretical and empirical evidence that SPON functions as a neural scaffolding, preventing representational drift and ensuring the integrity of latent distributions.

One can find the implementation at https://github.com/hxu105/SPON.

## 2. Spontaneous Neurons

In this section, we present **spontaneous neurons (SPON)**, a lightweight mechanism designed to complement *input activation sparsity* by stabilizing representations while preserving its efficiency benefits. We focus our analysis on

the decoder layers of transformer-based LLMs, where each layer consists of attention heads and MLP modules, and the forward pass of a layer can be expressed as a composition of linear mappings with non-linear operations.

### 2.1. Input Activation Sparsity

To formalize input activation sparsity, we introduce a sparsification operator $S(\cdot)$ applied to the input activations. For any linear transformation (e.g., in MLPs or Attentions), the sparse output is defined as

$$Y = \mathbf{W}S(X), \quad [S(X)]_i = \mathbb{1}_{\{|x_i|>\tau\}}x_i, \qquad (1)$$

where $\vec{w}_i$ is the $i$-th column of $\mathbf{W} \in \mathbb{R}^{k \times d}$, and $x_i$ is the corresponding input activation ($X \in \mathbb{R}^d$) and modulates the contribution of a specific feature direction $\vec{w}_i$, analogous to the firing strength of a neuron in biological systems. Input activation sparsity (Li et al., 2022; Luo et al., 2024; Storaï et al., 2025; Zhang et al., 2025) enforces input-dependent structure by selectively suppressing low-magnitude activations. $\mathbb{1}$ is an indicator function and $\tau$ is a threshold determined heuristically or in a data-driven manner. This operation induces sparsity directly in the hidden states, enabling the selective skipping of inactive feature directions and their associated weight columns, which can reduce memory access and computation, as illustrated in Figure 2.

Importantly, this mechanism does not permanently remove parameters nor modify the underlying activation functions and gating units; instead, it dynamically screens neuron activations conditioned on the input. Accordingly, activation sparsity can be viewed as a form of *input-dependent neuron deactivation*, which is efficient but may introduce representational instability when a large fraction of pretrained activations are suppressed. This observation motivates the introduction of spontaneous neurons as a stabilizing complement, which we detail next.

### 2.2. Baseline Firing Rates and Spontaneous Neurons

While activation sparsity is effective for reducing inference cost, it introduces a representational challenge. As spar-

sity increases, different inputs tend to activate increasingly weakly overlapping subsets of hidden dimensions, leading to unstable internal representations and a loss of shared semantic anchors across inputs (depicted in Figure 1).

Biological neural systems exhibit *spontaneous activity*, where neurons show baseline firing even in the absence of external stimuli, encoding prior expectations, stabilizing downstream responses, and providing a persistent reference against which stimulus-driven signals are interpreted. (Hubel & Wiesel, 1962; Arieli et al., 1996; Kenet et al., 2003; Fox & Raichle, 2007; Deco & Corbetta, 2011).

Motivated by this principle, we introduce *spontaneous neurons (SPON)*: a mechanism that injects a small set of learnable, input-independent activation vectors to each module. SPONs act as static representational scaffolds to complement dynamically selected neurons under activation sparsity and conceptually distill global, input-agnostic information from the dense model and reintroduce it as a stable prior, thereby reducing distributional drift induced by sparsity.

### 2.3. Details of Spontaneous Neurons

Concretely, for a sparsified linear transformation, we incorporate SPON into the forward pass by

$$Y = \mathbf{W}\,\mathbf{S}(X) + \mathbf{W}\,\vec{\alpha}, \tag{2}$$

where $\vec{\alpha}$ denotes a static spontaneous activation vector shared across inputs to a given layer. Since $\vec{\alpha}$ is independent of $X$, the term $\mathbf{W}\vec{\alpha}$ can be equivalently folded into a bias vector after training. As a result, SPON introduces no additional matrix multiplications and incurs zero inference-time overhead (Table 5 and Appendix D)

Let $\mathcal{A} := \{\vec{\alpha}_\ell\}$ denote the set of spontaneous activations. We learn solely these parameters by aligning the sparse model's output distribution with that of the dense model using a lightweight post-training calibration procedure. For a calibration input sequence $u \sim \mathcal{D}$, let $\mathbf{z}(u)$ and $\tilde{\mathbf{z}}(u; \mathcal{A})$ denote the output logits of the dense and sparse models respectively. We minimize the loss over $\mathcal{A}$:

$$\mathcal{L}(\mathcal{A}) = \mathbb{E}_{u \sim \mathcal{D}} \Big[ D_{\mathrm{KL}}\Big( \sigma\left(\mathbf{z}(u)\right) \,\|\, \sigma\left(\tilde{\mathbf{z}}(u; \mathcal{A})\right) \Big) \Big], \quad (3)$$

where $\sigma(\cdot)$ refers to the softmax function. This objective effectively distills global, input-agnostic information from the dense model into the SPON parameters, allowing them to act as a stable reference that complements the input-dependent active neurons.

Overall, SPONs provide a principled way to compensate for the representational distortion introduced by activation sparsity. By injecting stable baseline activations that can be absorbed into bias terms, they preserve the efficiency advantages of sparse inference while substantially improving accuracy and representation stability.

### 2.4. Theoretical Analysis: Fisher-Weighted Correction

To understand SPON further, we analyze the final linear projection layer, where the sparsity impact aggregates. Let $X$ denote the input hidden states. The dense and sparse logits are given by $\mathbf{z} = \mathbf{W}X$ and $\tilde{\mathbf{z}} = \mathbf{W}\mathbf{S}(X) + \mathbf{W}\vec{\alpha}$, respectively. We define the sparsity-induced residual as $\mathbf{e}(X) = \mathbf{W}X - \mathbf{W}\mathbf{S}(X)$. Minimizing the KL divergence yields the following first-order optimality condition (details in Appendix B.1):

$$\mathbb{E}_u \left[ \mathbf{W}^\top \mathbf{H}(\mathbf{W}\vec{\alpha} - \mathbf{e}(X)) \right] = 0, \tag{4}$$

where $\mathbf{H}$ is the Hessian of the loss with respect to its second logit argument, evaluated at $\mathbf{z}$, which coincides with the **Fisher Information Matrix** of the output distribution $\sigma(\mathbf{z})$.

This condition reveals that minimizing the KL divergence drives SPON to approximate the sparsity residual $\mathbf{e}(X)$ under a Fisher-weighted metric. The Hessian $\mathbf{H}$ acts as an importance matrix, guiding the optimization to align $\mathbf{W}\vec{\alpha}$ with the sparsity residual $\mathbf{e}(X)$ specifically along the directions where output probabilities are most sensitive (i.e., directions corresponding to large eigenvalues). This theoretical insight explains the efficacy of our approach: SPON mitigates the representational drift induced by sparsity, thereby enhancing overall representational stability.

## 3. Experiments and Analyses

In this section, we will discuss about our experiment settings, results, and their possible enlightenment for future designing and understanding LLMs.

### 3.1. Models, Datasets, & Baselines

We evaluate spontaneous neurons on three widely used LLMs—**Llama3-8B** (Dubey et al., 2024), **Qwen3-8B** (Yang et al., 2025), and **Mistral-7B** (Jiang et al., 2023)—and further include Llama3-1B to examine effects on smaller models. Wikitext (Merity et al., 2016) serves both as calibration data for training spontaneous neurons and as a benchmark for language modeling performance, measured by perplexity. To assess zero-shot generalization, we adopt six tasks: CommonsenseQA (Talmor et al., 2019), TruthfulQA (Lin et al., 2022), OpenBookQA (Mihaylov et al., 2018), MedM-CQA (Pal et al., 2022), MMLU (Hendrycks et al., 2021b;a), and MathQA (Amini et al., 2019). For all tasks, we utilize EleutherAI LM Harness (Gao et al., 2024) workflow and report mean performance. Details are in Appendix C.

We select TEAL (Liu et al., 2024) as the primary baseline, and we also include other SOTA baselines like LaRoSA (Liu et al., 2025a), WINA (Chen et al., 2025), R-Sparse (Zhang et al., 2025), and WAS (Wang et al., 2025) as described in section 3.4. As noted in Liu et al. (2024), TEAL applies activation sparsification to only 99% of tokens during the prefill

stage, which may cause more severe degradation when sparsifying the initial tokens due to attention sink (Xiao et al., 2024). In contrast, other SOTA methods apply sparsification uniformly across all tokens without special treatment. For a fair comparison, we implement full activation sparsification for TEAL as well as for our proposed SPON.

### 3.2. Spontaneous Neurons Improve Language Modeling

| Method | Sparsity | Language Modeling | | | |
|---|---|---|---|---|---|
| | | L-1B | L-8B | M-7B | Q-8B |
| Full | 0% | 12.14 | 6.75 | 5.49 | 8.99 |
| TEAL | 25% | - | 6.88 | 5.52 | 9.04 |
| SPON | 25% | - | 6.92 | 5.58 | 8.94 |
| TEAL | 50% | 17.03 | 8.34 | 6.00 | 9.75 |
| SPON | 50% | 15.43 | 7.83 | 5.86 | 9.26 |
| TEAL | 60% | - | 11.62 | 6.90 | 11.38 |
| SPON | 60% | - | 9.63 | 6.51 | 10.42 |

*Table 1.* Perplexity results on language modeling task. SPONs help to approximate the full dense model's performance across different sparsity levels and heterogeneous backbone models. L: Llama3, M: Mistral, and Q: Qwen3.

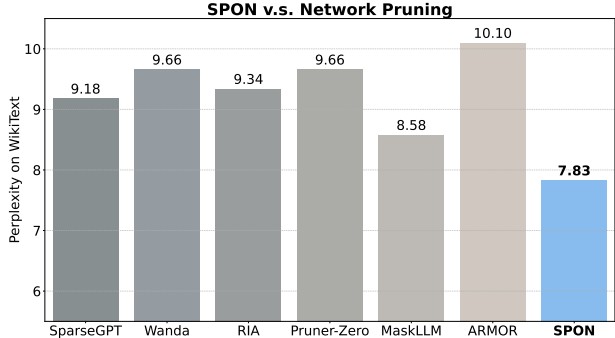

*Figure 3.* SPON outperforms SOTA network pruning methods.

Table 1 summarizes the language modeling performance under different levels of *activation sparsity*. At moderate sparsity (25%), SPON exhibits negligible perplexity degradation, consistent with prior observations that LLMs are inherently tolerant to partial activation masking (Liu et al., 2024). As sparsity increases to more aggressive regimes (50% and beyond), purely activation-based methods such as TEAL experience pronounced performance drops, reflecting accumulated representation error caused by aggressively zeroing input activations. In contrast, SPON consistently mitigates this degradation across all evaluated backbones.

To verify that SPON does not rely on using the same dataset for calibration and evaluation, we recalibrated SPON on C4 (Raffel et al., 2020) while evaluating on WikiText for Llama3-8B and Mistral-7B at 50% activation sparsity. The

resulting perplexities (**7.95** and **5.94**, respectively) remain superior to their baselines, demonstrating that SPON does not rely on dataset-specific alignment. SPON leaves pre-trained weights unchanged and instead compensates for the systematic activation-level shift introduced by sparsification, making the learned correction broadly transferable.

Finally, we make further comparisons between SPON and network pruning methods under 50% sparsity on Llama3-8B, including SparseGPT (Frantar & Alistarh, 2023), Wanda (Sun et al., 2023), RIA (Zhang et al., 2024a), Pruner-Zero (Dong et al., 2024), MaskLLM (Fang et al., 2024), and AR-MOR (Liu et al., 2025b). Unlike network pruning, SPON operates entirely in the activation space, preserves the original model parameters, and restores representational capacity, enabling highly sparse inference while maintaining performance comparable to dense models.

### 3.3. SPON Preserves Pretrained Knowledge

We evaluate SPON at 50% sparsity across a suite of zero-shot benchmarks spanning commonsense reasoning, mathematics, medical knowledge, and truthfulness, in order to assess its ability to preserve pretrained capabilities under aggressive activation sparsity. Figure 4 reports the *normalized mean accuracy* (the performance ratio between sparse and dense models), with comprehensive task-wise results provided in Appendix E.

Across all backbones, SPON consistently outperforms the TEAL baseline, demonstrating that spontaneous neurons effectively compensate for the representational voids introduced by sparsification. Remarkably, these improvements are achieved by injecting only a *single* spontaneous neuron per layer, corresponding to less than **0.016%** additional parameters and zero inference overhead. As formalized in Appendix A, activation sparsity with ratio $r$ is equivalent to dynamic neuron-level pruning that discards $r \times 100\%$ of weight columns during each forward pass. Consequently, at 50% sparsity, the model effectively utilizes only half of its parameters during inference, and the remaining performance gap largely stems from the fundamental information loss induced by aggressive parameter reduction rather than limitations of SPON itself.

More broadly, these findings challenge the common assumption that bias-like components are redundant in modern LLMs. Instead, they suggest that under activation sparsity, minimal input-independent activations can serve as effective *representational anchors* that stabilize hidden states and preserve generalization across architectures and task domains without sacrificing efficiency.

To further validate the scalability of this approach, we extended our evaluation to high-capacity models, comparing SPON and TEAL on Qwen3-32B and Llama3-70B.

*Figure 4.* Comparing TEAL and SPON on general and mathematical reasoning. We report normalized mean accuracy. Each point in the radar capability charts is (sparse performance)/(dense performance).

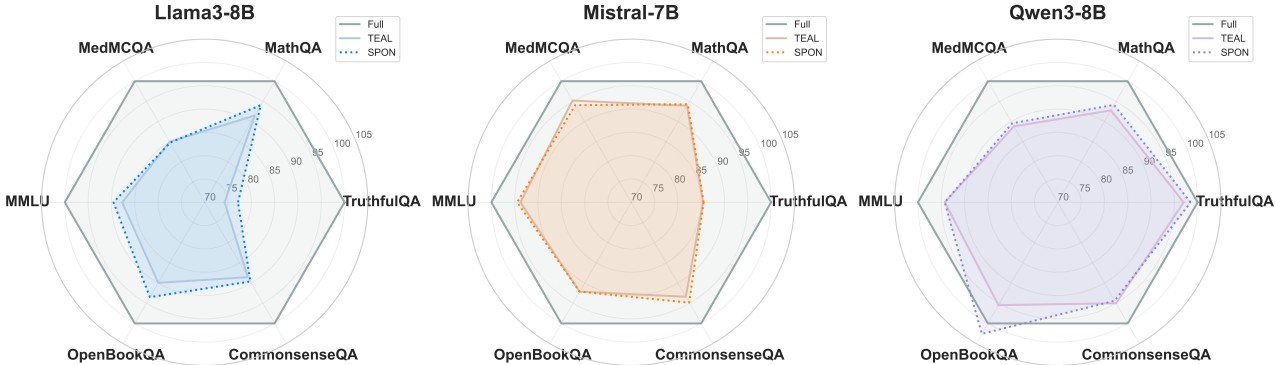

*Table 2.* SPON provides consistent improvements over LaRoSA, WINA, WAS, and R-Sparse. We emphasize that SPON is **orthogonal** to these methods, as SPON corrects the residual induced by activation masking. Thus, the methods could potentially be combined for further gains, reinforcing that SPON offers a general and complementary mechanism for improving sparse inference stability.

| Backbone LLM | Method | ARC-Challenge | ARC-Easy | BoolQ | PiQA | Winogrande | Average |
|---|---|---|---|---|---|---|---|
| | LaRoSA | 48.04% | 74.75% | 77.55% | **79.98%** | 68.98% | 69.82% |
| | WINA | 48.81% | 75.00% | 81.25% | 79.16% | 70.64% | 70.97% |
| Llama3-8B | R-sparse | 46.42% | 76.94% | 76.73% | 79.92% | 67.80% | 69.56% |
| | WAS | 47.06% | **78.83%** | 77.49% | 78.63% | 70.52% | 70.51% |
| | SPON | **51.11%** | 76.89% | **82.54%** | 78.40% | **70.88%** | **71.96%** |
| | LaRoSA | 49.06% | 75.55% | 80.92% | 81.61% | 69.93% | 71.41% |
| | WINA | 52.30% | 77.57% | 81.59% | 81.34% | 70.88% | 72.74% |
| Mistral-7B | R-sparse | 47.18% | 78.91% | 82.81% | 79.92% | **72.69%** | 72.30% |
| | WAS | 48.72% | 79.66% | 77.49% | 79.55% | 72.59% | 71.60% |
| | SPON | **56.31%** | **81.02** | **84.98%** | **82.05%** | 70.48% | **74.97%** |

As illustrated in Figure 5, the performance advantages of SPON persist at larger scales, yielding relative accuracy improvements of **0.75%** for Qwen3-32B and **0.96%** for Llama3-70B. Notably, these gains are achieved with less than **0.016%** additional parameters, as SPON is injected only into the down-projection layers of each MLP expert, while introducing zero inference overhead. This consistent upward trend across model sizes confirms that SPON is not merely a heuristic for small models, but a robust and practical mechanism for maintaining the integrity of large-scale sparse LLMs during deployment. Overall, these results suggest that SPON is not merely a heuristic for small models, but a scalable and practical solution for preserving representational integrity in sparse LLM deployment.

### 3.4. Comparison with SOTA Methods

To further contextualize the effectiveness of SPON, we compare against several recent extensions of TEAL, including LaRoSA (Liu et al., 2025a), WINA (Chen et al., 2025), R-Sparse (Zhang et al., 2025), and WAS (Wang et al., 2025), which enhance activation sparsity through orthog-

onal rotations, models weight information, singular-value refinements, and Bayesian priors, respectively. Table 2 summarizes the comparison on Winogrande (Sakaguchi et al., 2019), PiQA (Bisk et al., 2020), BoolQ (Clark et al., 2019), ARC (Clark et al., 2018) at 50% sparsity.

The results demonstrate that SPON consistently outperforms existing benchmarks across different model architectures, further reinforcing our claim of SPONs stabilizing sparse inference. We emphasize that SPON is *orthogonal* to these existing techniques as it uniquely focuses and addresses the representational drift induced by activation masking. Collectively, our empirical results across various LLM families show that SPON functions as a robust neural scaffolding, yielding a relative performance lift of 1.178% across a diverse suite of tasks.

### 3.5. Hidden Representation Analysis

Input activation sparsity masks a subset of activation entries for each token, inevitably inducing a distributional shift in hidden representations. We therefore examine whether

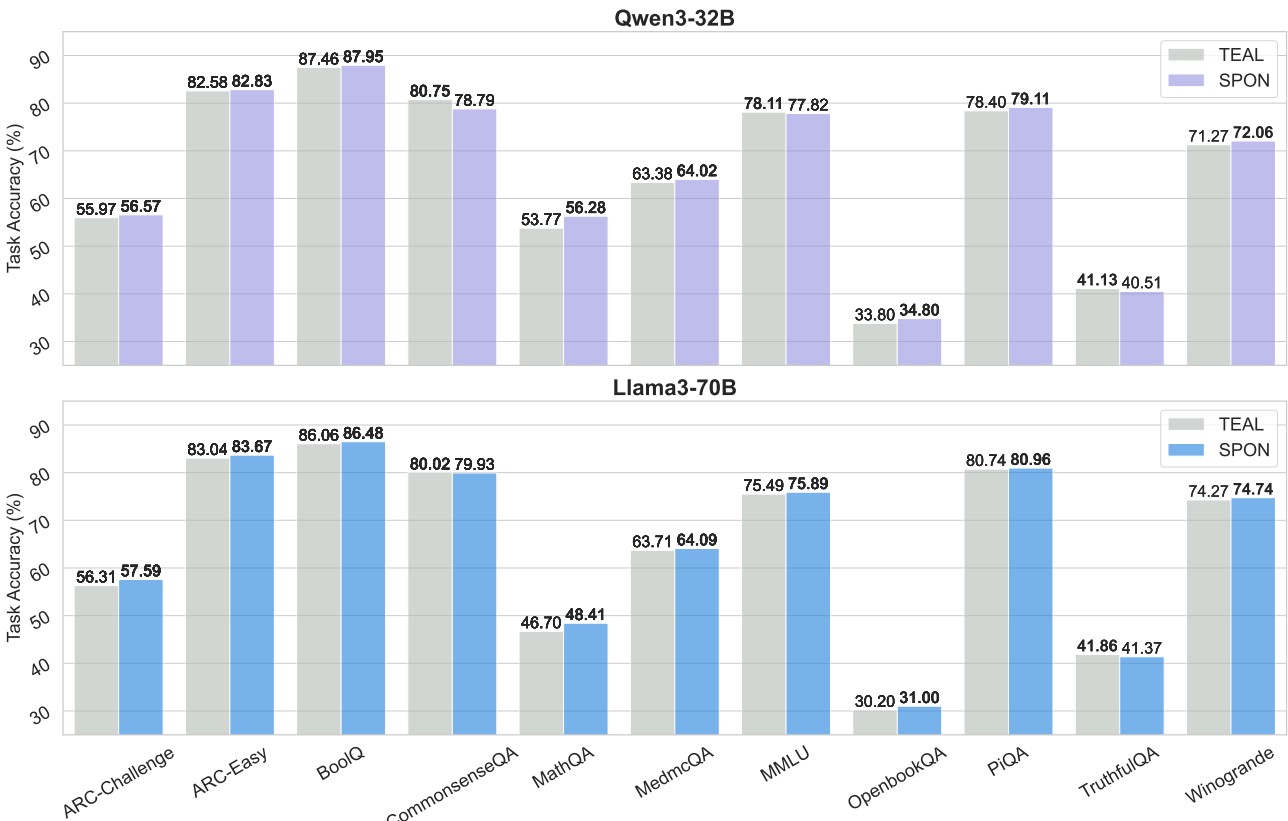

*Figure 5.* Scale-up experiments on Qwen3-32B and Llama3-70B. For 11 task from various domains, LLMs with SPON yield better performance in general. The activation sparsity ratio is set at 50%.

SPONs can mitigate this representational drift. Concretely, we randomly sample 50 prompts from the WikiText-raw-v2 test set (Merity et al., 2016), extract hidden states from (i) the dense model, (ii) the activation-sparse model (TEAL), and (iii) the activation-sparse model augmented with SPON, and compare their representations.

We quantify representational alignment using centered kernel alignment (CKA) (Kornblith et al., 2019) across decoder layers, and complement this with t-SNE visualizations (Maaten & Hinton, 2008) to illustrate geometric shifts in the latent space. In Figure 6, activation-sparse models with SPON exhibit consistently higher layerwise CKA similarity to the dense model than TEAL alone, indicating improved preservation of pretrained representations. This trend is further supported by Figure 6, where the average deviation $\ell_2$ in the latent space decreases from 8.69 (TEAL) to 8.21 (SPON) for Llama3-8B, with similar patterns observed in other backbones (Figure 13). Together, these results demonstrate that SPON substantially reduces the distributional shift introduced by activation sparsity, providing a mechanistic explanation for their improved knowledge retention. More details and visualizations are in Appendix F.

### 3.6. Inference Latency

To evaluate hardware efficiency, we benchmark end-to-end single-batch decoding latency following the TEAL (Liu et al., 2024) protocol. Using `GPT-Fast` with CUDA graphs and `torch.compile` enabled, we measure Llama3-8B and Mistral-7B at 50% uniform sparsity on a single NVIDIA A6000 GPU (300W power limit).

As shown in Table 3, SPON maintains the high efficiency of sparse inference, achieving speed-ups consistent with TEAL (up to $1.6\times$). Mechanistically, spontaneous neurons are implemented as additive bias terms within linear layers. Given that matrix multiplication dominates the computational cost, this marginal addition introduces no measurable latency penalty. Consequently, SPON provides enhanced representational modeling with negligible impact on throughput or memory footprint.

To further quantify efficiency, we report GFLOPs, GMACs, and parameter counts in Appendix D. SPON introduces no additional FLOPs or MACs during inference, as spontaneous activations are folded into fused bias terms after calibration. The resulting overhead is asymptotically negligible relative to GEMM computation and introduces effectively zero marginal latency in modern BLAS and Triton kernels,

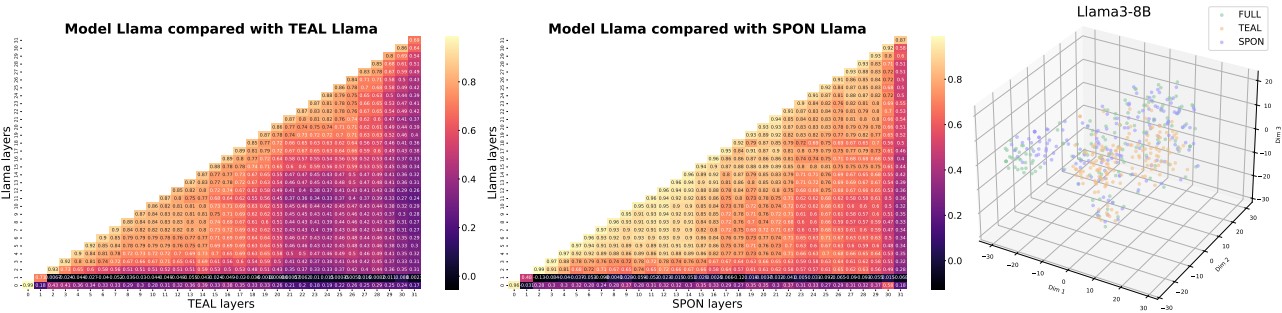

*Figure 6.* The CKA matrix for showing the representational similarity between activation-dense and activation-sparse feature spaces; the T-SNE visualization of hidden representation drifts, where SPON consistently exhibits relatively smaller shift. More visuals are F.

while adding only ∼0.016% parameters for Llama3-8B.

*Table 3.* Inference throughput (tokens/sec) at 50% sparsity. SPON preserves the efficiency gains of sparse computation.

| Method | Llama3-8B | Mistral-7B |
|---|---|---|
| Full (0%) | 42.65 | 44.17 |
| TEAL (50%) | 64.28 | 71.25 |
| SPON (50%) | 64.13 | 71.84 |

### 3.7. (Continual) Pretraining with SPON

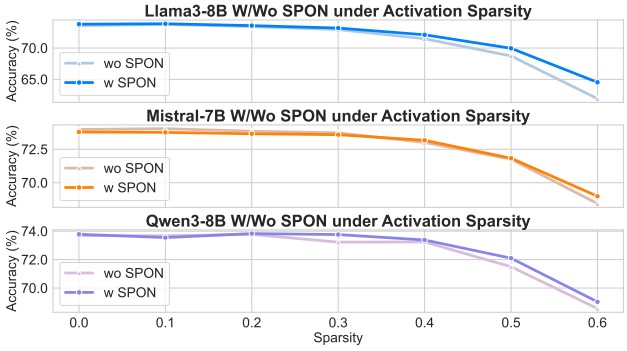

*Figure 7.* Pretraining LLMs with SPON outperforms standard LLMs under activation sparsity. Numerical results are in G.

Beyond post-training adaptation, incorporating SPON during pretraining enhances architectural robustness under activation sparsity. We simulate this by continuing next-token prediction training on WikiText, with SPON as the sole learnable parameters, and evaluate zero-shot generalization across Winogrande, PiQA, BoolQ, ARC, and MMLU.

As illustrated in Figure 7, SPON-integrated models consistently outperform standard baselines across all sparsity levels. This performance gap widens significantly at higher sparsity ratios, where standard models suffer catastrophic degradation. This trend, consistent across all evaluated backbones, suggests that SPON enables the internalization of stable representational priors that remain accessible even under aggressive input sparsification.

Collectively, these findings demonstrate that SPON provides a negligible-parameter mechanism to systematically improve model resilience. By fostering sparsity-awareness from the outset, SPON facilitates the development of LLMs that are robust to sparse computation by construction without compromising dense-model performance.

## 4. Ablation and Exploration Studies

We next conduct a series of ablation studies to isolate the key design factors behind SPON's effectiveness and offer some preliminary researches for potential broader impacts.

### 4.1. Necessity of Learning via $\vec{\alpha}$ Activation

*Table 4.* Learning from $\vec{b}$ only yields suboptimal results for L-1B.

| Learning from | # Neurons | Perplexity |
|---|---|---|
| | 1 | 17.07 |
| | 2 | 16.83 |
| $\vec{b}$ | 4 | 16.54 |
| | 8 | 16.43 |
| | 16 | 16.43 |
| $\vec{\alpha}$ | 1 | **15.43** |

Although spontaneous neurons can be algebraically absorbed into bias terms at inference time, learning $\vec{b}$ directly as biases is substantially less expressive, shown in Table 4. Empirically, replacing the spontaneous activation term $\mathbf{W}\vec{\alpha}$ with one bias $\vec{b}$, BitFit (Ben Zaken et al., 2022), or multiple biases using a self-ensemble method (Wortsman et al., 2021) with performance quickly saturating and remaining well below that of a single learned spontaneous activation.

This gap highlights a key distinction: learning $\vec{\alpha}$ in activation space constrains corrections to lie in the pretrained feature subspace spanned by $\mathbf{W}$, allowing gradients to align with and selectively compensate for information lost under sparsification. In contrast, bias-only learning lacks this representational alignment and therefore cannot effectively

recover missing semantics, underscoring the necessity of learning spontaneous neurons at the activation level rather than directly as biases. More details are in Appendix I.

## 4.2. Where to Inject Spontaneous Neurons

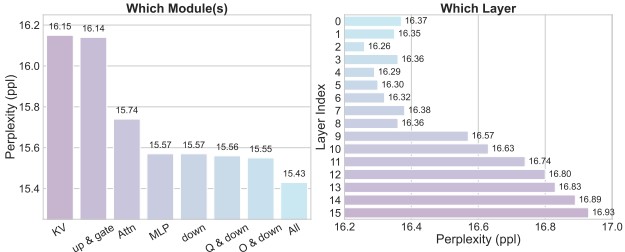

*Figure 8. Left*: We show that only injecting extra spontaneous neurons in down projection of MLP module can be comparable as injecting them to each block; *right*: We continue to investigate which layer favors more about the spontaneous neurons.

We next examine *where* spontaneous neurons are most effective. Using Llama3-1B on WikiText with perplexity as the metric, we evaluate injecting SPON into different components and layers. As shown in Figure 8 (*Left*), MLP blocks benefit substantially more than Attention blocks. In particular, adding SPON only to MLPs achieves a PPL of 15.57, compared to 15.74 when applied solely to Attentions. The *down projection* consistently yields the largest gains, matching the performance of more invasive configurations while using fewer additional parameters (0.0057% extra).

We further study layerwise sensitivity by injecting SPON into the MLP down projection of a single layer at a time. Figure 8 (*right*) shows that lower layers exhibit stronger improvements, consistent with prior evidence that bottom layers encode more localized and semantic representations (Dong et al., 2025; Sun et al., 2025; Sonkar & Baraniuk, 2023; He et al., 2024). Overall, these results indicate that SPONs are most effective when placed in early-stage MLP down projections, where they act as compact carriers of pretrained knowledge and provide stable priors that counteract the representational drift induced by activation sparsity.

## 4.3. Other Post-Training Approaches

Since SPON introduces a small number of additional parameters, we compare it to generic post-training techniques for restoring performance under activation sparsity. We evaluate LoRA (Hu et al., 2022) under the same calibration objective and evaluation protocol as in Section 3.7. As shown in Figure 9, increasing the rank, and thus parameter count, does not reliably improve performance in the sparse regime. In contrast, SPON yields larger and more stable gains while adding only **0.016%** parameters; even LoRA with rank 1 requires twice the parameter budget yet underperforms SPON. This suggests that recovering sparsity-induced degradation

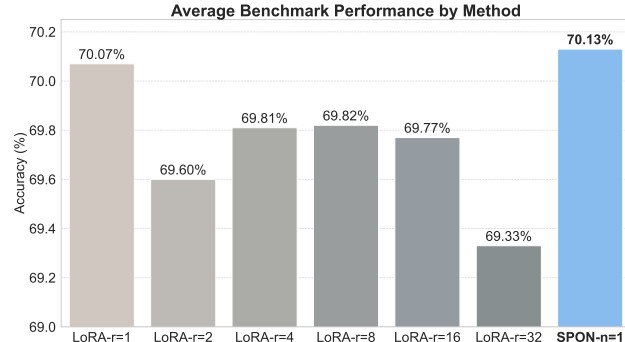

*Figure 9.* SPON achieves stronger performance with fewer parameters than LoRA. Task-wise numerical results are in Appendix H.

requires structural alignment with pretrained representations rather than generic low-rank adaptation.

We also compare against SCAP (Chua et al., 2024), a statistics-based post-training activation pruning scheme. SCAP sparsifies only MLP blocks at lower sparsity ratios (0.35, 0.35, and 0.55 for up, gate, and down projections, respectively) and reports a perplexity of *19.05* for Llama3-1B on WikiText, considerably worse than SPON's **15.43** at 0.5 sparsity ratio. Together, these results show that SPON is both more parameter-efficient and effective at stabilizing sparse computation than existing post-training approaches.

## 4.4. Harmony with MoE Structures

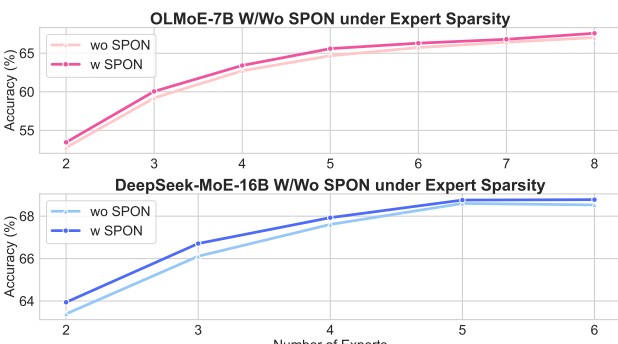

*Figure 10.* SPON improves performance under reduced expert budgets in MoE models, indicating that it complements sparse expert routing and supports more efficient MoE inference.

Thus far our experiments focused on dense LLMs. We now show that SPON also complements sparse *Mixture-of-Experts* (MoE) architectures. We follow the same training setup and evaluation protocol as in Section 3.7, but instead of varying activation sparsity, we impose *expert sparsity* by reducing the number of experts during inference. We evaluate two representative MoE models, **OLMoE-7B** (Muennighoff et al., 2024) and **DeepSeek-MoE-16B** (DeepSeek-AI, 2024). For MoE LLMs, SPON is only implemented on down projection for each expert. More details are in J.

Figure 10 reports performance as a function of the number of selected experts. Across both architectures, SPON consistently improves performance under reduced expert budgets, indicating that representational scaffolding even when sparse computation arises from expert routing rather than activation masking. This suggests that incorporating SPON during pretraining may reduce the number of experts required at deployment, enabling further sparsification of MoE inference while preserving generalization capabilities. By offering these preliminary observations, we hope to prompt further inquiry into the optimal synergy between expert routing and neural scaffolding provided by SPON.

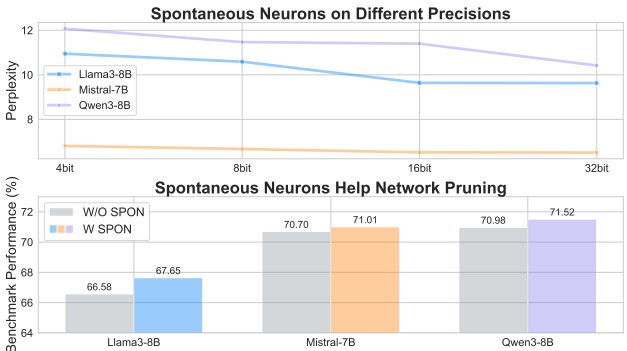

*Figure 11. Top*: Impact of quantization precision on spontaneous neurons and input sparsification. *Bottom*: LLMs with SPON yield better performance after network pruning.

### 4.5. Synergy with Quantization and Pruning

To evaluate the versatility of SPON as a foundational stabilizer, we examine its compatibility with weight quantization and network pruning—two critical pillars of efficient LLM inference. As shown in Figure 11 (top), SPON exhibits remarkable representational resilience across various quantization precisions, maintaining stable perplexity even at 60% sparsity levels. Notably, our architecture under `int4` quantization consistently outperforms the baseline results reported in the full 32-bit setting.

Furthermore, we observe that SPON provides a protective effect against network pruning. In Figure 11 (bottom) and Table 13, we apply the Wanda pruning method (Sun et al., 2023) to models continually pretrained with SPON. Following the assessment protocol in Section 3.7, we observe consistent performance gains across all evaluated backbones. These results indicate that SPON can also help recover the information density lost during weight-space pruning.

Collectively, these findings demonstrate that SPON opens new frontiers for extreme model compression. While the current implementation is inference-neutral, realizing the full hardware potential of these combined regimes requires the development of specialized kernels that jointly support quantization, pruning, and spontaneous activations, which

we leave as a direction for future research.

## 5. Related Work

**Efficient LLM Inference** for reducing the computational and memory cost of LLM inference has driven research on model compression, including unstructured and structured pruning (Frantar & Alistarh, 2023; Sun et al., 2023; Yin et al., 2023; Ma et al., 2023; Zhang et al., 2023; Xiao et al., 2024; Jiang et al., 2024; Zhang et al., 2024a; Dong et al., 2024; Fang et al., 2024; Liu et al., 2025b) and knowledge distillation (Hinton et al., 2015; Bick et al., 2024; Sreenivas et al., 2024). Pruning removes parameters or activations deemed redundant, while distillation transfers predictive behavior into compact forms.

**Activation Sparsity** has been exploited for efficient inference via alternative nonlinearities (Mirzadeh et al., 2023; Zhang et al., 2024b; So et al., 2021) and hardware-aware designs (Song et al., 2024; Alizadeh et al., 2024). Recent methods (Lee et al., 2024; Chua et al., 2024; Liu et al., 2024; Zhang et al., 2025; Liu et al., 2025a; Wang et al., 2025; Chen et al., 2025) enforce or leverage activation sparsity through training-free or lightly calibrated schemes. While effective for reducing computation, these methods often introduce representational drift. Our work complements this line by directly addressing the representational instability induced by activation sparsity, stabilizing hidden states while retaining full efficiency gains under activation sparsity.

## 6. Conclusion

We propose *Spontaneous Neurons (*SPON*)*, a lightweight mechanism that stabilizes activation-sparse large language models by mitigating representational drift. SPON anchors pretrained representations, compensates for sparsity-induced residuals, and consistently improves performance with zero inference overhead. Our theoretical analysis further provides a principled explanation of its effectiveness through Fisher-weighted correction. Together, these results suggest that spontaneity plays a fundamental role in robust sparse computation and establish SPON as a practical primitive for efficient LLM deployment.

More broadly, while modern LLM architectures increasingly discard components such as bias terms by default, our findings offer a new perspective on revisiting bias spontaneity at the activation level, motivating future architectures to rethink seemingly redundant design choices through the lens of representational stability and sparsity robustness.

## Impact Statement

This work focuses on improving the efficiency, stability, and context-handling capabilities of LLMs. The potential

positive societal impacts include: more efficient and stable training methods can lower the computational cost of developing and deploying LLMs, potentially making them more accessible to researchers and organizations with limited resources.

## Acknowledgments

This works was partially supported by the National Science Foundation (NSF) under Grant No. 2514002, SUNY-IBM AI Collaborative Research Alliance.

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

## A. Activation Sparsity as Neuron-Level Pruning

In this section, we first formalize the notation system used throughout the appendix and clarify the connection between activation sparsity and neuron-level pruning.

### A.1. General Notations & Preliminaries

Unless otherwise stated, we adopt the following notation for both Appendix A and Appendix B:

- $X \in \mathbb{R}^d$: input hidden states

- $S : \mathbb{R}^d \to \mathbb{R}^d$: input sparsification

- $\mathbf{W} \in \mathbb{R}^{k \times d}$: weight matrix of a given linear transformation (e.g., in MLPs or Attentions)

- $\vec{\alpha} \in \mathbb{R}^d$: spontaneous neuron vector

### A.2. Equivalence Mapping

We formalize the equivalence between activation sparsity and dynamic pruning to elucidate the underlying mechanism.

We define a *neuron* as a column vector of a learnable weight matrix $\mathbf{W} = [\vec{w}_1, \ldots, \vec{w}_d]$. The forward pass for an input vector $X$ can be expressed as a linear combination of neuron weight vectors:

$$\mathbf{y} = \mathbf{W}X = \sum_{i=1}^{d} x_i \vec{w}_i, \tag{5}$$

where $x_i$ denotes the activation of neuron $i$. Applying an activation sparsity function $S(\cdot)$, such as threshold masking, yields:

$$\tilde{\mathbf{y}} = \mathbf{W}S(X) = \sum_{i=1}^{d} \mathbb{1}_{\{|x_i|>\tau\}} x_i \vec{w}_i, \tag{6}$$

where $\mathbb{1}$ is the indicator function. From this perspective, activation sparsity performs *dynamic neuron pruning*: whenever $|x_i| \leq \tau$, neuron $i$ is pruned for that forward pass.

Importantly, since $S(\cdot)$ is independent of matrix multiplication, masking can be applied directly to $\mathbf{W}$ by zeroing out the corresponding columns, which effectively reduces the number of multiply-accumulate operations. The same reasoning extends to batched inputs $\mathbf{X} \in \mathbb{R}^{d \times m}$. In this case, activation sparsity allows the model to amortize efficiency gains across the entire sequence.

This equivalence highlights a key challenge discussed in the main text (Figure 1): because neurons are pruned on a per-token basis, the set of neurons that remain active across longer sequences rapidly shrinks. Even at modest sparsity levels (e.g., 10%), it becomes difficult to identify neurons that stay permanently active, underscoring the need for mechanisms such as SPON to provide stable representational support.

## B. Theoretical Analysis

In this section, we provide the detailed derivation of the first-order optimality condition presented in Section 2.4, verifying that SPON performs a Fisher-weighted error correction. We also discuss the implications of this mechanism on model complexity and generalization.

### B.1. Derivation of the Optimality Condition

Our theoretical analysis focuses on the *final linear projection layer* where the representational impact of activation sparsity aggregates into the output distribution. In addition to the base notations established in Appendix A.1, we further define the following terms for this specific derivation:

- $\mathbf{e}(X) = \mathbf{W}X - \mathbf{W}S(X)$: sparsity-induced residual

- $\mathbf{z} = \mathbf{W}X$: output logits of dense model

- $\tilde{\mathbf{z}} = \mathbf{W}S(X) + \mathbf{W}\vec{\alpha}$: output logits of sparse model

- $\sigma : \mathbb{R}^k \to (0,1)^k$: softmax function

- $\mathbf{p} = \sigma(\mathbf{z})$, $\mathbf{q} = \sigma(\tilde{\mathbf{z}})$: output distributions of dense and sparse models, respectively

Consider KL-divergence within the loss definition in Eq. 3 as a function of $\tilde{\mathbf{z}}$.

$$
\begin{aligned}
\mathcal{L} &= D_{\mathrm{KL}}\left(\sigma(\mathbf{z}) \,\|\, \sigma(\tilde{\mathbf{z}})\right) = D_{\mathrm{KL}}\left(\mathbf{p} \,\|\, \mathbf{q}\right) \\
&= \sum_i \mathbf{p}_i \log \frac{\mathbf{p}_i}{\mathbf{q}_i} = \sum_i \mathbf{p}_i \log \mathbf{p}_i - \sum_i \mathbf{p}_i \log \mathbf{q}_i
\end{aligned}
\tag{7}
$$

We can compute the first order derivative w.r.t. the second variable $\tilde{\mathbf{z}}$:

$$
\begin{aligned}
\nabla_{\tilde{\mathbf{z}}}\mathcal{L} &= \nabla_{\tilde{\mathbf{z}}}\left(\sum_i \mathbf{p}_i \log \mathbf{p}_i - \sum_i \mathbf{p}_i \log \mathbf{q}_i\right) \\
&= -\nabla_{\tilde{\mathbf{z}}}\left(\sum_i \mathbf{p}_i \log \frac{\exp(\tilde{\mathbf{z}}_i)}{\sum_j \exp(\tilde{\mathbf{z}}_j)}\right)_\ell \\
&= \left(-\sum_i \mathbf{p}_i \frac{\partial}{\partial \tilde{\mathbf{z}}_\ell} \log \frac{\exp(\tilde{\mathbf{z}}_i)}{\sum_j \exp(\tilde{\mathbf{z}}_j)}\right)_\ell \\
&= \left(-\sum_i \mathbf{p}_i \left(\mathbb{1}_{\{i=\ell\}} - \frac{\exp(\tilde{\mathbf{z}}_\ell)}{\sum_j \exp(\tilde{\mathbf{z}}_j)}\right)\right)_\ell \\
&= (-\mathbf{p}_\ell + \mathbf{q}_\ell)_\ell = \mathbf{q} - \mathbf{p}
\end{aligned}
\tag{8}
$$

Then, the second order derivative w.r.t. $\tilde{\mathbf{z}}$ is

$$
\begin{aligned}
\nabla_{\tilde{\mathbf{z}}}^2 \mathcal{L} &= \left[\frac{\partial}{\partial \tilde{\mathbf{z}}_j}(\mathbf{q}_i - \mathbf{p}_i)\right]_{ij} = \left[\frac{\partial}{\partial \tilde{\mathbf{z}}_j}\left(\frac{\exp(\tilde{\mathbf{z}}_i)}{\sum_\ell \exp(\tilde{\mathbf{z}}_\ell)}\right)\right]_{ij} \\
&= \left[\frac{\mathbb{1}_{\{i=j\}}\exp(\tilde{\mathbf{z}}_i)\sum_\ell \exp(\tilde{\mathbf{z}}_\ell) - \exp(\tilde{\mathbf{z}}_j)\exp(\tilde{\mathbf{z}}_i)}{\left(\sum_\ell \exp(\tilde{\mathbf{z}}_\ell)\right)^2}\right]_{ij} \\
&= \left[\mathbb{1}_{\{i=j\}}\frac{\exp(\tilde{\mathbf{z}}_j)}{\sum_\ell \exp(\tilde{\mathbf{z}}_\ell)} - \mathbf{q}_i \mathbf{q}_j\right]_{ij} \\
&= \mathrm{diag}(\mathbf{q}) - \mathbf{q}\mathbf{q}^\top =: \mathbf{H}(\tilde{\mathbf{z}})
\end{aligned}
\tag{9}
$$

It is clear that $\mathbf{H}$ is symmetric. Substituting $\tilde{\mathbf{z}} = \mathbf{z}$ yields $\mathbf{H}(\mathbf{z}) = \mathrm{diag}(\mathbf{p}) - \mathbf{p}\mathbf{p}^\top$.

Then, by Taylor expansion, we have

$$
\begin{aligned}
\mathcal{L}(\tilde{\mathbf{z}}) &= \mathcal{L}(\mathbf{z} - (\mathbf{e}(X) - \mathbf{W}\vec{\alpha})) \\
&\approx \mathcal{L}(\mathbf{z}) - (\mathbf{e}(X) - \mathbf{W}\vec{\alpha})^\top \nabla\mathcal{L}(\mathbf{z}) + \frac{1}{2}(\mathbf{e}(X) - \mathbf{W}\vec{\alpha})^\top \nabla^2\mathcal{L}(\mathbf{z})(\mathbf{e}(X) - \mathbf{W}\vec{\alpha})
\end{aligned}
\tag{10}
$$

The first term becomes zero since $D_{\mathrm{KL}}(\mathbf{p} \,\|\, \mathbf{p}) = 0$; and the second term becomes zero as $\nabla\mathcal{L}(\mathbf{z}) = \mathbf{p} - \mathbf{p} = 0$. Therefore, we have

$$
\mathcal{L}(\tilde{\mathbf{z}}) \approx \frac{1}{2}(\mathbf{e}(X) - \mathbf{W}\vec{\alpha})^\top \mathbf{H}(\mathbf{z})(\mathbf{e}(X) - \mathbf{W}\vec{\alpha})
\tag{11}
$$

Finally, we connect this to the Fisher Information. For a categorical distribution, the Fisher Information Matrix $\mathcal{I}(\mathbf{z})$ is exactly the Hessian of the negative log-likelihood, which coincides with $\mathbf{H}(\mathbf{z})$. Taking the expectation over the calibration dataset $u \sim \mathcal{D}$ and computing the gradient w.r.t. $\vec{\alpha}$ yields the first-order optimality condition:

$$
\nabla_{\vec{\alpha}}\mathbb{E}_u[\mathcal{L}] \approx \mathbb{E}_u\left[-\mathbf{W}^\top\mathbf{H}(\mathbf{z})(\mathbf{e}(X) - \mathbf{W}\vec{\alpha})\right] = \mathbb{E}_u\left[\mathbf{W}^\top\mathbf{H}(\mathbf{z})(\mathbf{W}\vec{\alpha} - \mathbf{e}(X))\right] = 0
\tag{12}
$$

which recovers Eq. 4 in the main text. $\qquad\square$

## B.2. Discussion on Generalization and Complexity

While the derivation in Appendix B.1 establishes the optimality of SPON, we now address potential concerns regarding model complexity and overfitting.

### 1. MINIMAL COMPLEXITY OVERHEAD.

Let $\mathcal{H}_0$ and $\mathcal{H}_{\text{SPON}}$ denote the hypothesis spaces of the standard sparse model and the model augmented with SPON, respectively:

$$\mathcal{H}_0 = \{X \mapsto \mathbf{W}\mathbf{S}(X)\} \quad \text{vs.} \quad \mathcal{H}_{\text{SPON}} = \{X \mapsto \mathbf{W}\mathbf{S}(X) + \mathbf{W}\vec{\alpha} \mid \vec{\alpha} \in \mathbb{R}^d\}.$$

Introducing $\vec{\alpha}$ adds only $d$ trainable parameters per module. In the context of LLMs where the projection dimension $d$ is large but the training data (tokens) $N$ is massive, this increase in VC-dimension is negligible. According to statistical learning theory, the generalization error bound is given by:

$$\mathcal{E}_{\text{gen}} \leq \mathcal{E}_{\text{train}} + \mathcal{O}\left(\sqrt{\frac{\text{complexity}}{N}}\right).$$

By minimizing the KL divergence, SPON significantly reduces the first term ($\mathcal{E}_{\text{train}}$) by aligning the sparse and dense distributions. Simultaneously, the complexity penalty term remains virtually unchanged compared to the dense baseline. Thus, SPON improves fidelity without introducing a risk of overfitting.

### 2. MITIGATING DISTRIBUTIONAL SHIFT.

Mechanistically, activation sparsity $S(\cdot)$ acts as a non-linear filter that can introduce a systematic shift in the mean of the hidden states, even if the original inputs were zero-centered. The spontaneous activation vector $\vec{\alpha}$ allows the model to explicitly learn and counteract this shift. Unlike a generic bias term $b$, learning $\vec{\alpha}$ in the activation space ensures that the correction $\mathbf{W}\vec{\alpha}$ lies strictly within the valid subspace of the pre-trained weights $\mathbf{W}$. This geometric constraint acts as a form of regularization, ensuring that the restored information is semantically consistent with the original model's feature space.

## C. Implementation Details

We use the Wikitext-raw-v2 dataset (Merity et al., 2016) as calibration data to train the spontaneous neurons. During training, each weight block $\mathbf{W}$ is associated with a spontaneous activation $\vec{\alpha}$ (a set of learnable parameters), and we transform these activations into bias terms inside $\mathbf{W}$, i.e., $\forall \mathbf{W}_l : \vec{b}_{\mathbf{W}_l} = \mathbf{W}_l\vec{\alpha}_l$. Once training is complete, the spontaneous activations $\vec{\alpha}$ become fixed, and treating them as bias terms (referred to as spontaneous neurons) introduces no additional matrix multiplication. This preserves the model's inference efficiency, as shown in Table 3.

We evaluate the effectiveness of spontaneous neurons on three main LLM backbones: the Llama3 family (Dubey et al., 2024), Mistral-7B (Jiang et al., 2023), and the Qwen3 family (Yang et al., 2025). All models share the same training configuration: a learning rate of 1e-5, 10 training epochs, a batch size of 8, and a block size of 128 for dataset chunking. We train all models on four Nvidia A6000 GPUs. For larger models such as Llama3-8B, Mistral-7B, and Qwen3-8B, the training typically takes between 1.14 and 2.25 hours (1.14h for Llama3-8B, 1.85h for Mistral-7B, and 2.25h for Qwen3-8B), while smaller models can be trained in under 20 minutes.

## D. Computational Efficiency and Parameter Analysis

To quantify the computational overhead of SPON, we report GFLOPs, GMACs, and total parameter counts in Table 5. Our analysis demonstrates that SPON maintains the efficiency of the underlying sparsity engine (e.g., TEAL), introducing no additional FLOPs or MACs during inference. This zero-cost overhead is achieved by folding the learned spontaneous activations into the linear layer's bias terms after calibration.

From a formal complexity perspective, a linear transformation $\mathbf{y} = \mathbf{x}\mathbf{W}^\top + \mathbf{b}$ theoretically incurs $Nk$ additional additions for the bias term. However, it is standard practice in LLM literature to report FLOPs based solely on the General Matrix Multiplication (GEMM) operations. This convention is justified by the fact that for an input $\mathbf{x} \in \mathbb{R}^{N \times d}$ and weight $\mathbf{W} \in \mathbb{R}^{k \times d}$, the ratio of bias additions to total floating-point operations is $\frac{Nk}{2Nkd} = \frac{1}{2d}$. As the feature dimension $d$ scales (e.g., $d = 4096$ for Llama3-8B), this ratio becomes asymptotically negligible.

Furthermore, modern BLAS (Blackford et al., 2002) and Triton kernels (Tillet et al., 2019) typically employ operator fusion, integrating bias addition into the register-level execution of the Fused Multiply-Add (FMA) instructions. Consequently, the marginal latency of the bias term is effectively zero. While the total parameter count increases slightly—by approximately 0.016% for Llama3-8B—due to the introduction of bias vectors in layers where they were previously absent, the operational intensity remains unchanged. Given that FLOPs serve as a proxy for computational work rather than a direct measure of wall-clock time, omitting the fused bias term provides the most accurate first-order approximation of the model's true inference budget.

*Table 5.* **Computational Complexity Analysis.** We compare GFLOPs, GMACs, and total parameter counts across different models. Note that while SPON slightly increases the parameter count by introducing bias vectors, it maintains identical FLOP/MAC counts to baselines.

| Method | Llama3-8B | Mistral-7B | Llama3-8B | Mistral-7B | Llama3-8B | Mistral-7B |
|---|---|---|---|---|---|---|
| | GFLOPs | | GMACs | | # Params | |
| Full (0%) | 1010.16 | 1078.34 | 505.06 | 539.15 | 8,030,261,248 | 7,248,023,552 |
| TEAL (50%) | 1010.16 | 1078.34 | 505.06 | 539.15 | 8,030,261,248 | 7,248,023,552 |
| SPON (50%) | 1010.16 | 1078.34 | 505.06 | 539.15 | **8,031,765,760** | **7,249,432,576** |

## E. Zero-Shot Table

We report the numerical results for the zero-shot experiment in Table 6.

*Table 6.* Numerical value for zero-shot performance of 3 backbone LLMs on 6 QA tasks. We use ⇑ to highlight the improve of SPON over TEAL.

| Model | Method | CommonsenseQA | MathQA | MedMCQA | MMLU | OpenBookQA | TruthfulQA |
|---|---|---|---|---|---|---|---|
| | FULL | 77.23% | 39.46% | 58.93% | 68.05% | 33.80% | 54.03% |
| Llama3-8b | TEAL | 68.39% | 36.11% | 50.08% | 59.69% | 30.40% | 40.14% |
| | SPON | 69.21% ⇑ | 37.12% ⇑ | 49.96% | 60.95% ⇑ | 31.60% ⇑ | 41.69% ⇑ |
| | FULL | 66.91% | 36.92% | 46.31% | 59.05% | 35.60% | 66.82% |
| Mistral-7B | TEAL | 62.49% | 34.67% | 44.08% | 55.37% | 32.80% | 57.08% |
| | SPON | 63.47% ⇑ | 34.81% ⇑ | 43.56% | 55.78% ⇑ | 32.80% | 57.15% ⇑ |
| | FULL | 78.71% | 49.61% | 59.62% | 72.97% | 31.00% | 54.41% |
| Qwen3-8b | TEAL | 74.77% | 46.03% | 52.95% | 68.78% | 29.60% | 52.78% |
| | SPON | 74.28% | 46.70% ⇑ | 53.41% ⇑ | 68.79% ⇑ | 31.80% ⇑ | 53.59% ⇑ |

## F. Latent Analysis

To further investigate whether model with SPON semantically correspond to layers of the original dense model, we employ Centered Kernel Alignment (CKA), a widely established metric for comparing representational structures (Kornblith et al., 2019; Kriegeskorte et al., 2008). Given two sets of representations $X$ and $Y$ (e.g., activations from different models or layers), CKA is defined as:

$$\text{CKA}(K, L) = \frac{\text{HSIC}(K, L)}{\sqrt{\text{HSIC}(K, K)\text{HSIC}(L, L)}} \tag{13}$$

where $K = XX^\top$ and $L = YY^\top$ denote the Gram matrices (using a linear kernel) and HSIC is Hilbert-Schmidt Independence Criterion (Gretton et al., 2005). We employ a minibatch implementation with an unbiased estimator of HSIC (Davari et al., 2022) and evaluate on the WikiText.

Figure 12 visualize the similarity heatmaps for self, TEAL, and SPON comparisons. We observe that for activation sparse model with SPON, the diagonal entries reach over 95% similarity to the activation dense model while the TEAL counterpart often lose the resemblance of baselines. Even for off-diagonal entries, SPON expresses greater correspondence to those in self-CKA heatmap than that for TEAL, meaning that adding SPON does recover the information flow inside the language model and make the model representations more stable and robust under activation sparsity.

In Figure 13, we aim to empirically verify that adding a spontaneous neuron can avoid such an distributional mismatch. To validate it, we conducted the following steps: i) we randomly select 50 prompts from wikitext-raw-v2 test set (Merity et al., 2016) (each prompt may have 64 token maximally, which results in more than 1000 tokens in end) and extract the hidden representation within the full LLMs and TEAL sparse one. ii) Subsequently, we performed the same operation on the sparse LLM with spontaneous neurons. iii) Finally, we used t-SNE (Maaten & Hinton, 2008) to visualize the hidden representation distributions. Furthermore, we quantify the deviation of scatter point distributions and numerically exhibit the distributional shifts in Figure 13.

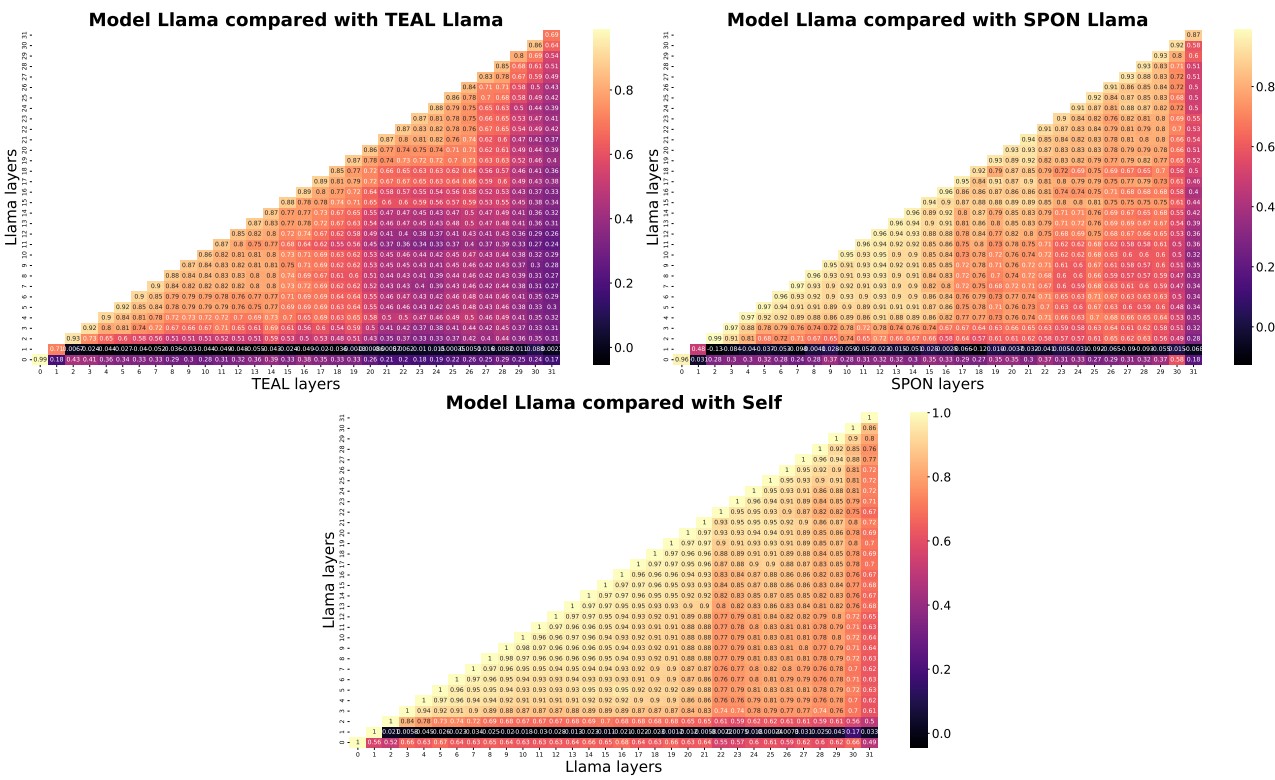

*Figure 12.* CKA.

## G. Continual Pretraining with SPON

While SPON can be applied post-training, its intended use is as a pretraining-time component. Due to compute and data constraints, we simulate this setting via continual pretraining on WikiText, using standard next-token cross-entropy without activation sparsity during training. We then evaluate robustness under activation sparsity after training.

As shown in Tables 7–9, models continually pretrained with SPON exhibit consistently improved robustness across backbones, particularly at higher sparsity levels (50% and 60%). This indicates that SPON encourages the model to internalize stable representational baselines during pretraining, making sparsified inference more resilient. These results support the claim that SPON is beneficial not only for post-training calibration but also as a lightweight pretraining-time mechanism for building sparsity-aware LLMs.

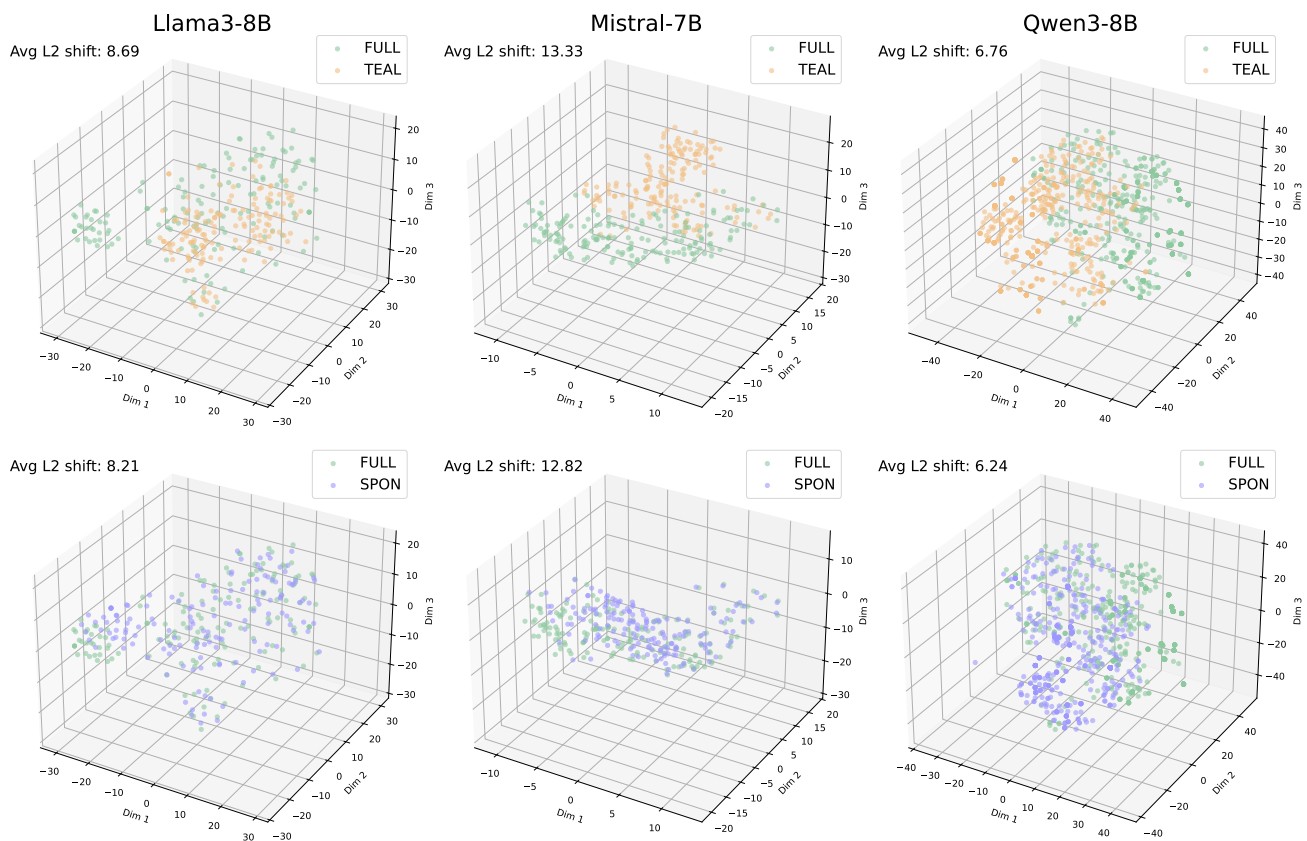

*Figure 13.* The T-SNE visualization of all tested LLM backbones.

*Table 7.* Llama3-8B

| Sparsity | SPON | ARC-Challenge | ARC-Easy | BoolQ | MMLU | PiQA | Winogrande |
|---|---|---|---|---|---|---|---|
| 0.0 | ✗ | 55.03% | 79.63% | 84.07% | 68.05% | 80.96% | 73.72% |
|  | ✓ | 55.55% ⇑ | 79.84% ⇑ | 84.10% ⇑ | 68.09% ⇑ | 81.66% ⇑ | 73.64% |
| 0.1 | ✗ | 55.46% | 79.80% | 84.19% | 68.05% | 81.07% | 74.11% |
|  | ✓ | 55.63% ⇑ | 79.84% ⇑ | 84.25% ⇑ | 68.14% ⇑ | 81.61% ⇑ | 73.80% |
| 0.2 | ✗ | 55.29% | 79.55% | 84.01% | 67.50% | 81.07% | 73.09% |
|  | ✓ | 55.38% ⇑ | 79.67% ⇑ | 84.40% ⇑ | 67.63% ⇑ | 81.07% | 73.32% ⇑ |
| 0.3 | ✗ | 54.52% | 79.00% | 83.67% | 66.75% | 80.36% | 73.40% |
|  | ✓ | 54.95% ⇑ | 78.96% | 83.94% ⇑ | 67.11% ⇑ | 81.01% ⇑ | 73.16% |
| 0.4 | ✗ | 53.50% | 78.16% | 84.22% | 64.44% | 78.56% | 70.01% |
|  | ✓ | 54.69% ⇑ | 78.03% | 83.36% | 64.78% ⇑ | 80.74% ⇑ | 71.11% ⇑ |
| 0.5 | ✗ | 47.87% | 75.17% | 82.57% | 59.69% | 77.64% | 69.53% |
|  | ✓ | 52.13% ⇑ | 76.64% ⇑ | 81.87% | 60.66% ⇑ | 79.22% ⇑ | 69.22% |
| 0.6 | ✗ | 40.78% | 69.07% | 76.88% | 48.58% | 73.39% | 62.83% |
|  | ✓ | 44.71% ⇑ | 71.13% ⇑ | 78.96% ⇑ | 52.15% ⇑ | 74.86% ⇑ | 65.35% ⇑ |

*Table 8.* Mistral-7B

| Sparsity | SPON | ARC-Challenge | ARC-Easy | BoolQ | MMLU | PiQA | Winogrande |
|----------|------|---------------|----------|-------|------|------|------------|
| 0.0 | ✗ | 58.87% | 82.62% | 85.84% | 59.73% | 82.64% | 74.19% |
|     | ✓ | 58.96% ⇑ | 82.58% | 84.92% | 59.64% | 82.86% ⇑ | 73.80% |
| 0.1 | ✗ | 59.13% | 82.79% | 85.93% | 59.75% | 82.54% | 74.11% |
|     | ✓ | 58.87% | 82.66% | 84.86% | 59.70% | 82.81% ⇑ | 73.64% |
| 0.2 | ✗ | 58.79% | 82.79% | 85.96% | 59.59% | 82.81% | 73.16% |
|     | ✓ | 59.13% ⇑ | 82.74% | 84.92% | 59.56% | 82.70% | 72.93% |
| 0.3 | ✗ | 58.45% | 82.91% | 86.39% | 59.02% | 82.26% | 73.32% |
|     | ✓ | 59.64% ⇑ | 82.58% | 85.05% | 59.00% | 83.08% ⇑ | 72.22% |
| 0.4 | ✗ | 57.42% | 81.73% | 85.72% | 57.95% | 82.81% | 72.45% |
|     | ✓ | 59.13% ⇑ | 81.82% ⇑ | 84.92% | 58.05% ⇑ | 83.03% ⇑ | 72.06% |
| 0.5 | ✗ | 55.12% | 80.60% | 84.89% | 56.20% | 81.56% | 72.22% |
|     | ✓ | 56.31% ⇑ | 81.02% ⇑ | 84.98% ⇑ | 56.16% | 82.05% ⇑ | 70.48% |
| 0.6 | ✗ | 49.66% | 76.35% | 84.46% | 50.91% | 79.38% | 69.85% |
|     | ✓ | 51.45% ⇑ | 77.78% ⇑ | 84.50% ⇑ | 51.82% ⇑ | 80.85% ⇑ | 67.56% |

*Table 9.* Qwen3-8B

| Sparsity | SPON | ARC-Challenge | ARC-Easy | BoolQ | MMLU | PiQA | Winogrande |
|----------|------|---------------|----------|-------|------|------|------------|
| 0.0 | ✗ | 56.40% | 80.93% | 86.61% | 72.97% | 77.69% | 67.56% |
|     | ✓ | 57.08% ⇑ | 80.51% | 86.42% | 72.97% | 77.37% | 68.35% ⇑ |
| 0.1 | ✗ | 56.31% | 81.06% | 86.82% | 72.87% | 77.53% | 67.48% |
|     | ✓ | 56.91% ⇑ | 80.39% | 86.42% | 72.89% ⇑ | 77.48% | 67.25% |
| 0.2 | ✗ | 57.17% | 80.60% | 86.70% | 72.70% | 77.64% | 67.80% |
|     | ✓ | 57.25% ⇑ | 80.60% | 86.21% | 72.77% ⇑ | 77.31% | 68.82% ⇑ |
| 0.3 | ✗ | 56.14% | 80.39% | 86.97% | 72.07% | 76.71% | 67.09% |
|     | ✓ | 57.51% ⇑ | 80.64% ⇑ | 86.61% | 72.12% ⇑ | 76.88% ⇑ | 68.82% ⇑ |
| 0.4 | ✗ | 56.74% | 80.47% | 86.76% | 70.77% | 76.61% | 68.11% |
|     | ✓ | 57.42% ⇑ | 80.05% | 86.27% | 70.89% ⇑ | 77.04% ⇑ | 68.59% ⇑ |
| 0.5 | ✗ | 53.33% | 79.76% | 86.21% | 68.78% | 75.73% | 65.19% |
|     | ✓ | 55.29% ⇑ | 79.25% | 84.86% | 68.74% | 75.79% ⇑ | 68.67% ⇑ |
| 0.6 | ✗ | 48.98% | 76.98% | 84.19% | 62.48% | 74.86% | 63.85% |
|     | ✓ | 51.71% ⇑ | 77.31% ⇑ | 84.19% | 62.76% ⇑ | 74.37% | 63.85% |

## H. LoRA vs. SPON

LoRA and SPON are related in that both introduce auxiliary parameters that compensate for the residual error induced by activation sparsity. However, the two differ fundamentally in how this compensation is achieved. LoRA augments the weight space through learnable low-rank matrices $\mathbf{A}$ and $\mathbf{B}$, providing input-dependent corrections tailored for downstream adaptation. In contrast, SPON posits that sparsity-induced residuals can be addressed by lightweight, input-independent activation vectors that serve as stable representational baselines.

Empirically, Table 10 shows that increasing the LoRA rank—thereby increasing expressiveness and parameter count—does

not reliably preserve generalization under activation sparsity. SPON, despite introducing significantly fewer parameters, yields stronger and more stable performance in the sparse regime. These results suggest that recovering sparsity-induced degradation benefits more from structural alignment with pretrained activation patterns than from generic low-rank adaptation to the weight space.

*Table 10.* LoRA v.s. SPON on Llama3-8B

| Method | Setting | ARC-Challenge | ARC-Easy | BoolQ | MMLU | PiQA | Winogrande | Extra Params |
|--------|---------|---------------|----------|-------|------|------|------------|--------------|
| LoRA | rank=1 | 50.68% | 76.47% | **82.72%** | 60.54% | 78.13% | **71.90%** | 0.0326% |
| LoRA | rank=2 | 50.77% | 76.64% | 81.71% | 60.39% | 78.94% | 69.14% | 0.0652% |
| LoRA | rank=4 | **51.88%** | 76.14% | 82.57% | **61.14%** | 78.18% | 68.98% | 0.1303% |
| LoRA | rank=8 | 50.68% | 76.22% | 82.26% | 60.48% | **79.27%** | 70.01% | 0.2601% |
| LoRA | rank=16 | 49.57% | 76.64% | 81.96% | 60.59% | 78.29% | 71.59% | 0.5182% |
| LoRA | rank=32 | 50.09% | 75.84% | 81.68% | 60.28% | 79.05% | 69.06% | 1.0282% |
| SPON | #neuron=1 | 51.11% | **76.89%** | 82.54% | 60.95% | 78.40% | 70.88% | **0.0155%** |

## I. Details of Learning from $\alpha$ vs. $b$

In Section 4.1, we compared learning spontaneous activations $\alpha$ against learning bias terms $b$, highlighting the advantage of SPON over bias-only approaches such as BitFit (Ben Zaken et al., 2022). We provide experimental details here for completeness.

For the BitFit baselines, the *single-b* setting corresponds to finetuning only the bias term with zero initialization (since standard LLM implementations do not include bias terms, the initialization is implicitly zero). For the *multi-b* setting, we adopt a self-ensemble approach inspired by Wortsman et al. (2021): multiple bias vectors $\{b_i\}$ are trained jointly and encouraged to span a low-loss region by imposing pairwise orthogonality constraints. During training, a bias vector is sampled from the complex defined by the endpoints in the ensemble. Consistent with prior observations (Wortsman et al., 2021) and Table 4, increasing the number of endpoints improves performance.

However, even with self-ensembling and additional degrees of freedom, BitFit remains significantly weaker than SPON. This empirically supports our claim that compensating sparsity-induced degradation requires learning from activation space ($\alpha$) rather than solely adjusting bias terms ($b$), as activations provide richer alignment signals with pretrained representations.

## J. MoE Preliminary Studies

Mixture-of-Experts (MoE) models reduce inference cost by activating only a subset of experts per token. To examine whether SPON can benefit MoE-style sparsity, we study an *expert sparsity* setting analogous to activation sparsity in dense LLMs: we vary the number of active experts at inference, thereby reducing computation and parameter usage. As expected, continually shrinking the number of activated experts induces performance degradation in MoE LLMs.

Tables 11 and 12 show that MoE models (continually) pretrained with SPON (on WikiText, 1 epoch, 1e-6 learning rate) exhibit consistently better performance across different expert budgets. This suggests that a small number of input-independent spontaneous neurons can offset the loss of representational capacity induced by expert sparsity, effectively substituting for expensive expert computations at inference. These observations highlight the potential of SPON as a complementary mechanism for efficient MoE deployment. However, for more advanced implementation like how to combine with routing at pretraining stage, we would like to leave it for future studies.

*Table 11.* OLMoE

| #Experts | SPON | ARC-Challenge | ARC-Easy | BoolQ | MMLU | PiQA | Winogrande |
|---|---|---|---|---|---|---|---|
| 2 | ✗ | 32.94% | 56.36% | 65.41% | 39.03% | 67.79% | 55.09% |
|   | ✓ | 33.62% ⇑ | 58.84% ⇑ | 65.23% | 38.19% | 70.13% ⇑ | 54.70% |
| 3 | ✗ | 41.55% | 64.69% | 71.16% | 45.12% | 73.94% | 58.72% |
|   | ✓ | 42.06% ⇑ | 66.33% ⇑ | 71.31% ⇑ | 45.21% ⇑ | 75.08% ⇑ | 60.30% ⇑ |
| 4 | ✗ | 43.34% | 70.08% | 73.43% | 49.39% | 75.90% | 64.17% |
|   | ✓ | 44.45% ⇑ | 71.42% ⇑ | 73.85% ⇑ | 49.15% | 77.04% ⇑ | 64.56% ⇑ |
| 5 | ✗ | 46.08% | 72.01% | 75.63% | 51.64% | 76.61% | 66.06% |
|   | ✓ | 48.21% ⇑ | 73.27% ⇑ | 76.57% ⇑ | 51.66% ⇑ | 77.69% ⇑ | 66.14% ⇑ |
| 6 | ✗ | 48.46% | 73.57% | 76.76% | 52.39% | 77.69% | 65.51% |
|   | ✓ | 48.63% ⇑ | 74.66% ⇑ | 77.46% ⇑ | 52.63% ⇑ | 78.45% ⇑ | 65.98% ⇑ |
| 7 | ✗ | 48.98% | 74.66% | 76.91% | 52.91% | 78.51% | 66.54% |
|   | ✓ | 48.72% | 75.72% ⇑ | 77.74% ⇑ | 52.79% | 79.27% ⇑ | 66.54% |
| 8 | ✗ | 48.89% | 76.30% | 76.61% | 53.49% | 79.11% | 67.80% |
|   | ✓ | 50.68% ⇑ | 76.60% ⇑ | 77.46% ⇑ | 53.33% | 79.00% | 68.43% ⇑ |

*Table 12.* DeepSeek-MoE

| #Experts | SPON | ARC-Challenge | ARC-Easy | BoolQ | MMLU | PiQA | Winogrande |
|---|---|---|---|---|---|---|---|
| 2 | ✗ | 41.72% | 70.79% | 74.59% | 47.39% | 78.13% | 67.72% |
|   | ✓ | 43.69% ⇑ | 70.83% ⇑ | 75.11% ⇑ | 47.76% ⇑ | 78.51% ⇑ | 67.72% |
| 3 | ✗ | 45.73% | 74.03% | 76.54% | 52.04% | 79.65% | 68.67% |
|   | ✓ | 47.10% ⇑ | 74.03% | 77.25% ⇑ | 52.19% ⇑ | 79.92% ⇑ | 69.77% ⇑ |
| 4 | ✗ | 47.10% | 75.34% | 78.35% | 54.05% | 80.20% | 70.64% |
|   | ✓ | 47.95% ⇑ | 75.63% ⇑ | 78.65% ⇑ | 54.04% | 80.14% | 71.19% ⇑ |
| 5 | ✗ | 48.81% | 76.30% | 79.51% | 54.51% | 80.63% | 71.82% |
|   | ✓ | 49.23% ⇑ | 76.64% ⇑ | 80.21% ⇑ | 54.54% ⇑ | 80.25% | 71.67% |
| 6 | ✗ | 48.72% | 76.30% | 79.94% | 54.71% | 80.41% | 71.11% |
|   | ✓ | 48.98% ⇑ | 76.30% | 79.91% | 54.79% ⇑ | 80.36% | 72.38% ⇑ |

# K. SPON Helps Pruning

We additionally study whether SPON benefits model pruning, which introduces parameter-level sparsity rather than activation-level sparsity. To this end, we apply Wanda (Sun et al., 2023) pruning in Table 13 to LLMs (continually) pretrained with and without SPON, and evaluate them under the same protocol as Section 3.7. Across all backbones, models pretrained with SPON consistently achieve higher accuracy after pruning, indicating that spontaneous neurons help recover information lost due to weight corruption. These results further demonstrate that SPON improves robustness to multiple forms of sparsification and increases the practical applicability of sparse LLMs.

*Table 13.* Numerical value for after-pruning results on LLMs without SPON and with SPON.

| Model | SPON | ARC-Challenge | ARC-Easy | BoolQ | MMLU | PiQA | Winogrande |
|-------|------|---------------|----------|-------|------|------|------------|
| Llama3-8b | ✗ | 46.59% | 69.70% | 81.25% | 54.85% | 76.77% | 70.32% |
|           | ✓ | 48.12% ⇑ | 71.51% ⇑ | 81.77% ⇑ | 56.71% ⇑ | 77.69% ⇑ | 70.09% |
| Mistral-7B | ✗ | 52.05% | 80.64% | 85.81% | 54.42% | 80.30% | 70.96% |
|            | ✓ | 54.10% ⇑ | 81.14% ⇑ | 84.65% | 55.02% ⇑ | 80.52% ⇑ | 70.64% |
| Qwen3-8b | ✗ | 52.30% | 80.35% | 85.17% | 65.40% | 74.10% | 68.59% |
|          | ✓ | 53.58% ⇑ | 80.98% ⇑ | 84.28% | 66.07% ⇑ | 75.14% ⇑ | 69.06% ⇑ |

