# OpenReview forum: "Resting Neurons, Active Insights: Robustifying Activation Sparsity in LLMs via Spontaneity"
_ICML.cc/2026/Conference — ICML 2026 regular_

### Official Review · Reviewer_ZWkh · 2026-03-09

**Soundness:** 2
**Presentation:** 3
**Significance:** 3
**Originality:** 2
**Overall Recommendation:** 3
**Confidence:** 3

**Summary:**

The paper addresses the performance degradation of Large Language Models (LLMs) when subjected to high levels of activation sparsity. The authors hypothesize that this failure is due to "representational instability," where the dynamic suppression of neurons disrupts the stable latent anchors learned during pretraining. To mitigate this, they introduce Spontaneous Neurons (SPON), a biologically inspired mechanism that injects a small set of learnable, input-independent activation vectors into each module. These vectors act as "representational scaffolds" and are trained via a lightweight post-training calibration process to align the sparse model's output distribution with the original dense model. Crucially, at inference time, these vectors can be folded into existing bias terms, resulting in zero additional FLOPs or latency. The method is evaluated across multiple LLM backbones (Llama3, Mistral, Qwen3) and shows consistent improvements in language modeling and zero-shot reasoning tasks under high sparsity regimes.

**Compliance With Llm Reviewing Policy:**

Affirmed.

**Key Questions For Authors:**

1.	You mentioned that injecting only a single spontaneous neuron per layer is sufficient. Did you experiment with increasing the number of spontaneous neurons ($n > 1$)? If so, did performance saturate immediately, or was there a point of diminishing returns?

2.	In Figure 10, you show compatibility with 4-bit quantization. Does the scale of the spontaneous activation vector $\vec{\alpha}$ need to be adjusted differently for quantized weights compared to FP16?

3.	How much calibration data is required for $\vec{\alpha}$ to converge? Is it possible to learn effective spontaneous neurons using only a few dozen examples (e.g., few-shot calibration)?

4.	You claim SPON is inference-neutral because it's folded into bias. However, many high-performance kernels are "bias-free." Have you benchmarked the throughput on hardware that specifically does not have optimized bias-addition paths for GEMM?

**Strengths And Weaknesses:**

Strengths
The paper is well-motivated by biological analogies and provides a mathematically grounded approach to a practical problem in LLM efficiency. The proposed SPON mechanism is elegant in its simplicity, particularly the fact that it incurs zero inference overhead by utilizing bias folding. The empirical evaluation is extensive, spanning multiple model families and scales, and is supported by insightful latent representation analysis such as CKA similarity and t-SNE visualizations.
Weaknesses

•	Limited Theoretical Novelty: While the Fisher-weighted correction analysis is sound, the core idea of using learned bias terms to compensate for pruning/sparsity is not entirely groundbreaking.

•	Risk of Distributional Overfitting: The reliance on WikiText for calibration raises questions about domain specificity. While the authors provide some cross-dataset validation (C4 to WikiText), a truly rigorous examination of out-of-distribution (OOD) stability—essential for LLMs—is somewhat lacking.

•	Hardware Implementation Realities: The claim of "zero overhead" assumes that re-introducing bias vectors to every layer is trivial. However, many high-performance GEMM kernels in current LLM frameworks are optimized specifically for bias-free weights. Adding these back could theoretically interfere with operator fusion or memory alignment in ways the paper glides over.

•	Sequence Length Scalability: Figure 1 shows that commonly activated neurons decay exponentially with sequence length. However, the experiments are conducted on relatively short block sizes (128). It is unclear if SPON's effectiveness holds for long-context windows where the representational drift might be more non-linear and complex.

---

> ### Author Rebuttal · Authors · 2026-03-25
>
> We sincerely thank reviewer ZWkh for the constructive review and recognizing the practical value of our approach. We now address each concern in turn.
>
> **W1**: While SPON can be folded into a bias term at inference, it is fundamentally different from conventional bias tuning. We respectfully argue that the novelty of SPON operates on multiple dimensions:
>
> - The *identification* of representational drift as the root cause of activation-sparsity degradation is itself a novel scientific contribution, to our knowledge, no prior work has characterized this failure mode through the lens of distributional shift in hidden states, nor has it been addressed as a representational alignment problem.
>
> - SPON learns corrections in *activation space* rather than in *weight/bias space*. $\mathbf{W}\vec{\alpha}$ lies strictly within the pretrained feature subspace spanned by $\mathbf{W}$, constraining corrections to be consistent with the original representations. A plain bias $\vec{b}$ lacks this structural alignment; even $16\times$ more bias parameters (Table 4) cannot match a single SPON neuron. Our Fisher-weighted optimality analysis (Section 2.4) shows that SPON approximates the sparsity-induced residual $e(X) = \mathbf{W}X - \mathbf{W}S(X)$ along directions of highest output sensitivity, a property not shared by conventional bias tuning.
>
> Together, the combination of a novel problem formulation, a biologically-inspired mechanism with a rigorous theoretical foundation, and empirical superiority over bias-centric baselines constitutes meaningful and substantive novelty beyond the surface similarity to learned biases.
>
> **W2**: In Figure 4, Figure 6, Figure 9, Figure 11, and Table 2, models are calibrated on WikiText yet evaluated on tasks with no domain overlaps, consistently demonstrating out-of-distribution stability. Plus, the KL divergence objective (Eq. 3) distill from the dense model into the SPON parameters, anchoring to the prior knowledge of dense model rather than to any domain-specific signal. This is further studied in Section 3.7 and Section 4.4, where even replacing KL divergence with next-token prediction still yields consistent SPON improvements over the baseline. We therefore believe that the concern of distributional overfitting is not substantiated by the evidence in our submission.
>
> **W3 & Q4**: High-performance GEMM implementations (e.g., cuBLASLt, CUTLASS) utilize *fused epilogues* to incorporate bias addition at the register level before writing results to memory, bias fusion is therefore a fundamental feature of these kernels, not a source of interference. While "bias-free" might be the current default, "GEMM+Bias" is a well-supported and common primitive in HPC. Moreover, since bias vectors are $\mathcal{O}(n)$ while matrix products are $\mathcal{O}(n^{ω>2})$, the memory bandwidth cost is marginal. Empirically, Table 3 shows that SPON achieves throughput virtually identical to the TEAL efficiency, and Table 5 confirms zero additional FLOPs and MACs. These results are obtained using the same sparse GEMV kernel introduced in the TEAL paper. We will explicitly specify the kernel implementation details in Section 3.6 in the revision for clarity.
>
> **W4**: Long-Context is **NOT** the study question. While we do not claim any effectiveness of SPON over long-context exploration, we can analyze the long-context exploration using RoPE and SPON-induced bias. Let $\mathcal{R}\_{m}$ and $\mathcal{R}\_{n}$ be the matrix forms of RoPEs at $m$ and $n$ positions for a query-key pair $(q\_m, k\_n)$, we have $q\_{m}^{\top}\mathcal{R}\_{m}^{\top}\mathcal{R}\_{n}k\_{n}\to (q\_{m}+b\_{q})^{\top}\mathcal{R}\_{m}^{\top}\mathcal{R}\_{n}(k\_{n}+b\_{k})$ with SPON-induced bases $b\_{q}$ and $b\_{k}$. Given zero expectations of $\langle b\_{q}, k\_{n} \rangle$ and $\langle b\_{k}, q\_{m} \rangle$ in high-dimensional statistics, one can easily find it approximately recovers the GAU's relative position bias functions, which can also deduce the Sandwich RoPE design. We validate this by testing Llama3-8B w/wo SPON on LongBenchV2. SPON can reach average **34.46%** slightly better than 32.99% from the non-SPON counterpart.
>
> **Q1**. While we did not extensively benchmark $n>1$ neurons per layer, as minimal parameterization was a central design goal, we expect further gains from the self-ensemble trick discussed in Section 4.1, consistent with the benefits reported in the cited work and our own experiments on $\vec{b}$.
>
> **Q2**. No scale adjustment is needed.
>
> **Q3**. We did not formally establish a lower bound on the calibration data required for convergence, but we witness that calibrating with 44.8k WikiText datapoints yields comparable performance as that with 2.3M C4 samples. Moreover, we observe that a single epoch of calibration (even on WikiText) is sufficient to yield strong performance improvements across all tested backbones. Few-shot calibration of SPON is an interesting direction for future work exploration.

---

### Official Review · Reviewer_iBU3 · 2026-03-10

**Soundness:** 3
**Presentation:** 3
**Significance:** 3
**Originality:** 3
**Overall Recommendation:** 4
**Confidence:** 4

**Summary:**

To address the severe performance degradation of large language models caused by unstable representations during activation sparsification, this paper proposes a spontaneous neuron (SPON) mechanism inspired by biological neural systems. The method injects a small number of input-independent learnable activation vectors into the model to serve as stable representation anchors for sparse computation. These vectors are trained to match the distribution of dense models and can be fully absorbed into the bias terms after training, achieving zero extra inference overhead. Experiments demonstrate that SPON effectively stabilizes hidden-layer representations and significantly restores the performance of various large models under sparse inference without sacrificing efficiency.

**Compliance With Llm Reviewing Policy:**

Affirmed.

**Final Justification:**

I believe the paper's idea is simple and effective, and the authors conducted extensive experiments to validate the method's effectiveness; the method works very well. The authors' rebuttals addressed my main concerns and reinforced my previous recommendation for acceptance; therefore, I recommend this paper for acceptance.

**Key Questions For Authors:**

1. Could the authors explain the questions regarding the novelty of the method in the weaknesses?

2. Could the authors explain the questions regarding the accuracy comparison of the SPON method in the section on weaknesses?

In my opinion, I have not found any significant weaknesses in this paper. On the contrary, its strengths are more obvious. Therefore, I recommend accepting this paper.

**Limitations:**

N.A. The authors did not discuss the limitations and potential negative societal impacts of this work. However, to the best of my knowledge, this paper does not have any significant potential limitations or notable negative societal impacts.

**Strengths And Weaknesses:**

**Strengths**

1. Elegant and Practical Approach. The proposed SPON method is remarkably concise. By introducing input-independent “spontaneous neurons” to anchor hidden states and absorbing them into bias terms during inference, it significantly improves the performance of sparse models without introducing any extra computation or memory bandwidth overhead (zero inference overhead).

2. Solid Theoretical Grounding. The paper not only presents empirical results but also provides rigorous theoretical analysis in Section 2.4 and the appendix. The authors prove that the calibration process minimizing the KL divergence is essentially equivalent to a first-order optimal correction for the errors induced by sparsification, weighted by the Fisher information matrix. This provides strong mathematical support for the effectiveness of the method.

3. Comprehensive Empirical Validation. The authors conduct extensive experiments on multiple mainstream architectures (Llama3, Mistral, Qwen3, with model sizes ranging from 1B to 70B). The experiments cover not only language modeling (perplexity) and zero-shot downstream tasks but also investigate the compatibility of SPON with other orthogonal techniques such as MoE architectures, weight quantization, and network pruning, demonstrating strong generality and extensibility.

**Weaknesses**
1. The authors have packaged their method using a rich array of biological concepts, such as "spontaneous neural activity" and "baseline firing rate". However, from the mathematical and engineering essence, SPON is merely adding a learnable bias vector $\vec{\alpha}$ to the existing linear projection layer (i.e., $Y = W S(X) + W \vec{\alpha}$), fine-tuning it on the calibration set via KL divergence, and finally multiplying it with the weights and absorbing it into the original bias term. This approach is technically ingenious in engineering practice, but its theoretical innovation is relatively limited, and the forced application of a biological narrative appears somewhat far-fetched. Certainly, this simple yet effective approach is appreciable. I only hope the authors can clarify the novelty of their method.

2. Although SPON significantly outperforms baseline methods such as TEAL at 50% sparsity, the experimental results in Table 6 show that its zero-shot accuracy still has a non-negligible gap compared with the original full dense model. For instance, the accuracy of Llama3-8B on MMLU drops from 68.05% to 60.95%. In practical LLM deployment scenarios, the absolute degradation of core model capabilities is often highly sensitive. Sacrificing several percentage points of reasoning and commonsense performance for around 1.5× speedup may be unacceptable in certain high-demand applications.

3. Compared with TEAL, the accuracy improvement of SPON is relatively limited. For example, according to the data in Figure 11, the relative accuracy improvement is only 0.75% for Qwen3-32B and 0.96% for Llama3-70B. As shown in Table 6, the average accuracy gains across six downstream datasets are less than 1% for LLaMA3-8B, Mistral-7B, and Qwen3-8B.

4. I am more concerned with the comparison of average zero-shot accuracy on downstream datasets. I suggest adding a column for average zero-shot accuracy in Table 6, Figure 11, and other relevant tables and figures.

---

> ### Author Rebuttal · Authors · 2026-03-27
>
> We sincerely thank reviewer iBU3 for the thoughtful feedback and positive assessment. We hope the following responses address the concerns of our paper.
>
> **W1 & Q1**. We acknowledge that bias addition and bias-only fine-tuning techniques such as BitFit exist in the literature. However, we respectfully argue that SPON's novelty stems from multiple dimensions beyond the implementation of $\mathbf{W}\vec{\alpha}$ itself.
>
>
> - We are the first, to our knowledge, to identify and characterize the decay of commonly activated neurons across tokens (Figure 1) as a fundamental failure mode of activation sparsity. This finding motivates the entire framework and has not been presented or discussed in prior work.
> - We reframe activation sparsity as a *representational alignment problem* and develop SPON as a neural scaffolding mechanism that anchors dense model prior knowledge in *activation space* rather than weight/bias space. This distinction is non-trivial: learning $\vec{\alpha}$ such that $\mathbf{W}\vec{\alpha}$ lies within the pretrained feature subspace spanned by $\mathbf{W}$ geometrically constrains the correction to remain semantically consistent with the original model, a property that plain bias learning (BitFit) lacks, which is precisely why even $16\times$ more bias parameters cannot match a single spontaneous neuron (Table 4).
>
> - Theoretical grounding. The Fisher-weighted optimality analysis (Section 2.4) provides a principled justification showing that SPON approximates the sparsity residual $e(X) = \mathbf{W}X - \mathbf{W}S(X)$ along the directions of highest output sensitivity, distinguishing it from prior bias-tuning approaches.
>
> - Biological motivation. The spontaneous neural activity analogy is not merely decorative; it provides a conceptual framework that directly motivates the input-independence of ${\alpha}$ as a stable representational prior, analogous to baseline firing rates in biological systems. We will clarify the distinction between biological motivation and technical contribution more explicitly in the revision.
>
> **W2 & W3 & Q2**. We agree that the gap between sparse and dense models remains non-trivial at aggressive sparsity ratios, and we share the reviewer's view that closing this gap is an important direction for future research. However, we would like to contextualize this gap. As formalized in Appendix A, activation sparsity at ratio $r$ is equivalent to dynamic neuron-level pruning that discards $r \cdot 100$% of weight columns per forward pass. At 50% sparsity, the model effectively utilizes only half of its parameters during inference, while SPON contributes only 0.016% additional parameters. The performance gap is therefore largely attributable to the fundamental information loss from aggressive parameter reduction, not a deficiency of SPON itself. Notably, methods with far more parameters, such as LoRA (up to 1.03% extra parameters, Table 10) and multi-bias BitFit with self-ensemble, achieve strictly lower performance than a single spontaneous neuron, further underscoring SPON's parameter efficiency. The same reasoning applies at larger scales (Llama3-70B, Qwen3-32B), where SPON continues to provide consistent gains despite the inherent constraints of extreme sparsification.
>
> We acknowledge that the relative gains over TEAL are modest at larger model scales (0.75% for Qwen3-32B, 0.96% for Llama3-70B). We would note, however, that these improvements are achieved with even less than 0.016\% additional parameters as we only inject SPON in down projection inside each MLP expert and zero inference overhead, and that they are consistent across all tested architectures, scales, and task domains. The diminishing gap at larger scales is consistent with the known observation that larger models are inherently more robust to sparsification, leaving less room for correction. We will add a discussion of this scaling behavior in the revision.
>
> **W4**. We thank the reviewer for this constructive suggestion and will add average accuracy columns to Table 6, Figure 11, and all relevant tables and figures in the revision.

---

> > ### Author Rebuttal · Reviewer_iBU3 · 2026-04-01
> >
> > The authors have adequately addressed my concerns. I find the idea presented in this paper simple yet effective, and I appreciate such straightforward but impactful approaches. The authors have conducted thorough and comprehensive experiments to validate the effectiveness of their algorithm, demonstrating strong performance. I agree with the authors’ clarification regarding novelty. Although there remains a considerable accuracy gap compared with dense models, SPON contributes to narrowing this gap and achieves further improvements over state-of-the-art methods. I accept the authors’ clarification on the performance concerns of SPON. In summary, I believe this paper is worthy of acceptance and meets the acceptance criteria of ICML. I maintain my positive score and recommend acceptance. Good Luck!

---

> > > ### Author Response · Authors · 2026-04-01
> > >
> > > We sincerely thank reviewer iBU3 for the encouraging acknowledgement and continued support. We are glad our clarifications were satisfactory.
> > >
> > > Given the positive assessment and recommendation for acceptance, we would like to humbly inquire whether the reviewer would consider raising the score.​​​​​​​​​

---

### Official Review · Reviewer_UMFw · 2026-03-11

**Soundness:** 4
**Presentation:** 3
**Significance:** 3
**Originality:** 3
**Overall Recommendation:** 5
**Confidence:** 4

**Summary:**

The paper introduces a novel method for handling activation sparsity in neural networks, exploring applications in LLMs. The authors theorize that inducing sparsity in pre-trained networks causes harmful representational drift, and propose to solve this drift using a new technique they term "spontaneous neurons" (SPON). This technique adds a few (as little as one) activation vectors that are matrix multiplied with linear layer weight matrices and added on to the matmul of sparsified activations with the same weight matrix. After training, the activation vectors can be combined with the weight matrix and absorbed into a bias term, allowing for sparsity to retain computational benefits in the linear layer matmuls. The authors choose TEAL (a training-free method for activation sparsity) as a primary baseline, and demonstrate broad equivalence or superiority in language modeling, and general and mathematical reasoning, across a few representative architectures. They also find that SPON - an activation sparsity method - outperforms related weight pruning methods for achieving sparsity. They then compare to other more sophisticated methods for inducing activation sparsity, and again find broad equivalence or superiority, noting that their method is also orthogonal and could be combined with these methods for further gains (though this is not demonstrated). Next, they find broadly stronger representational similarity throughout the layers of Llama and its SPON variant, compared to Llama and its TEAL variant, in line with the notion the SPON helps prevent representational drift. SPON produces equivalent latency increases to TEAL relative to the base model, since both TEAL and SPON are tested at an equivalent sparsity level; however SPON performs generally better. The authors perform some preliminary analyses to demonstrate that SPON can be helpful for continual pretraining, and sparsifying MoE models. Preliminary analyses also suggest that spontaneous neurons may not be needed at every layer, but perhaps are most important in the down projection of the MLP module, providing a path to further reductions in parameter count for the method. Similarly, SPON performs at least as well as LoRA, using less parameters, though this is not tested extensively. Last, the authors suggest that SPON can be useful for lower precision inference, as well as network pruning. Overall, the authors extensively test their method and find decent gains across the board. Critically, the authors test a simple baseline that most readers will be wondering about far earlier than it is presented, which is replacing SPON with a trainable bias (a previously introduced method). Interestingly, SPON outperforms this method, even though it is mathematically equivalent after training; the prior implicit in the weight matrix may help SPON achieve better spontaneous activity than directly learning the activity, and the authors present compelling empirical evidence in favor of this notion.

**Compliance With Llm Reviewing Policy:**

Affirmed.

**Key Questions For Authors:**

1. Can you elaborate on the trade-off between accuracy, sparsity, and inference time? I believe this is a critical point that could use more air time. One idea is a colored scatter plot, with accuracy and inference time on x and y axes, and color indicating sparsity (or some other method for plotting all three parameters on a single graph).

2. Can you explain why TEAL was chosen as the primary baseline, compared to the other baselines that are compared less extensively?

3. Can you speak to the possible efficiency gains of LoRA? I was least impressed by the comparison of SPON to LoRA. The added parameter count is not particularly significant. If SPON allows for increased inference efficiency due to its sparsity, relative to LoRA, this should be highlighted.

4. Would there be any benefit to SPON in a pure pre-training setting?

5. Why do you call them spontaneous neurons, when a neuron would be a unit, whereas these neurons are vectors in activation space? I think spontaneous directions would be more accurate, and you can still keep the name SPON, or use SPOND (no need to credit me if you changeit).

**Limitations:**

No. There is no proper limitations section. The authors need to make some space for a discussion section. It's great to have a lot of results. It's not great to have no space for discussion.

**Strengths And Weaknesses:**

Soundness: the paper is very technically sound. The method is clever, is theoretically motivated, and is well supported by extensive empirical analyses.

Presentation: the paper is enjoyable to read, and is clearly presented. I commend the authors on a well written paper. I do think they may benefit from cutting down on some of the results in the main paper (and moving the less critical ones to the appendix) in order to provide more space for discussion, as the discussion is very minimal if not entirely absent. I would also argue that the title is misleading, as this paper does not perform any interpretability; even though sparsity might have the benefit of greater interpretability, it is not a focus of the paper and "active insights" should thus be removed from the title, in my opinion.

Significance: the paper addresses an important problem. They demonstrate improved accuracy relative to other sparsification methods, and also demonstrate the significance of this, achieving inference speed-ups. The latter point could be strengthened and/or clarified, which I will discuss below.

Originality: I think the method is quite original and clever, even if it is a relatively simple modification of prior methods. Even if the simplicity of the methodological contribution is considered negative, the extensive empirical analyses make up it. And I would argue this simplicity may be seen as a virtue.

---

> ### Author Rebuttal · Authors · 2026-03-29
>
> We sincerely thank reviewer UMFw for the careful and generous review. It is gratifying to see the core contributions recognized, and we hope the following responses address the remaining questions.
>
> **Q1**. We agree to reviewer UMFw's advice that figuring the relation between inference efficiency and sparsity can make our work more appealing, and we will include a figure for it in the revised paper. To provide a glance of this change, we offer a simple table of token per second measurements on Llama3-8B with different sparsity here.
> | Sparsity | Tokens/sec |
> |--|--|
> | 0.1 | 38.58 |
> | 0.2 | 41.45 |
> | 0.3 | 48.28 |
> | 0.4 | 48.07 |
> | 0.5 | 54.52 |
> | 0.6 | 66.67 |
> | 0.7 | 79.15 |
> | 0.8 | 90.29 |
> | 0.9 | 85.82 |
>
> For reference, Llama3-8B without SPON achieves 56.60 tokens/sec at 50% sparsity under identical conditions. Minor numerical differences from Table 3 arise from prompt variation, but the overall trend: faster inference at higher sparsity, with no latency penalty from SPON, is consistent with findings in TEAL and LaRoSA.
>
> **Q2**. TEAL serves as the foundational reference point for activation sparsity research in the LLM community: it establishes the core sparsification protocol (magnitude-based thresholding applied to input activations) that subsequent methods: LaRoSA (layerwise orthogonal rotation), WINA (weight-informed neuron activation), R-Sparse (rank-aware sparsity), and WAS (Bayesian scheduling), all build upon as extended variants. To verify methodological orthogonality, we test WINA w/wo SPON on language modeling below.
> |Model|SPON|PPL|
> |-|-|-|
> |Llama3-8B|✓|**7.94**|
> |Llama3-8B|✗|8.34|
>
> Selecting TEAL as the primary baseline allows us to isolate and evaluate SPON's contribution in the most controlled setting, while the comparisons in Table 2 against all four extensions demonstrate that SPON remains competitive across a broad range of sparsification strategies. We also note that this choice emphasizes a broader point: a concise and principled approach like SPON can be as effective as considerably more complex methods.
>
>
> **Q3**. We appreciate this observation and agree it deserves more emphasis. The inference efficiency advantage of SPON over LoRA is substantial and goes beyond raw parameter count. As formalized in Appendix A, activation sparsity is equivalent to dynamic neuron-level pruning: at sparsity ratio r, only (1-r)% of weight columns are active per forward pass, which is the direct source of the wall-clock speed-up. Thus, the efficiency gains is more about activation sparsity rather than the post-training pipeline. We will highlight this in the revision, and we do hope reviewer UMFw can explain this question with more details, so that we can come up more and deeper discussions.
>
> To put the parameter difference in perspective: LoRA at rank 32 introduces approximately 80M additional parameters and is comparable in scale to BERT-base, while SPON introduces approximately 1.2M parameters, even less than the size of TinyBERT. Despite this $62\times$ parameter advantage for LoRA, SPON consistently outperforms it (Table 10), which we attribute to the input-agnostic nature of $\vec{\alpha}$ being better suited for recovering a stable representational prior than input-dependent weight-space adaptation. We will add a dedicated discussion of this efficiency distinction in the revision.
>
> **Q4**. Yes, we would expect meaningful benefits. When an LLM is pretrained with SPON from the outset, the spontaneous neurons can internalize stable representational baselines as part of the training objective, potentially yielding a model that is robust to activation sparsity by construction rather than by post-hoc correction. Our continual pretraining experiments in Section 3.7 and Section 4.4 provide preliminary evidence: models trained with SPON consistently outperform standard baselines under both activation sparsity and expert sparsity, with the performance gap widening at more aggressive sparsity levels. We hypothesize that pretraining with SPON may also improve parameter utilization efficiency more broadly, as the spontaneous neurons may encourage the model to encode knowledge in a more sparsity-resilient manner. We consider full pretraining with SPON an important direction for future work.
>
> **Q5**. We thank the reviewer for this precise and thought-provoking observation, and we would like to share our views on naming SPON. Our use of "neurons" is motivated by the formalization in Appendix A, where we define a neuron as a column vector $\vec{w}_i$​ of a weight matrix $\mathbf{W}$, and activation sparsity as the dynamic pruning of such columns. Under this convention, $\vec{\alpha}$ acts as a static activation pattern over these neurons, so the final $\vec{b}=\mathbf{W}\vec{\alpha}$ is a spontaneous neuron. That said, we acknowledge the ambiguity and will revisit the terminology, including the possibility of adopting "spontaneous directions" or a related term, in the revision, while retaining the SPON acronym.

---

> > ### Author Rebuttal · Reviewer_UMFw · 2026-04-04
> >
> > The authors resolved all of my concerns. I remain supportive of the paper. Congrats on the nice work.

---

> > > ### Author Response · Authors · 2026-04-04
> > >
> > > Thank you for your positive review! We’re glad to find you appreciate the work!

---

### Official Review · Reviewer_b6pM · 2026-03-12

**Soundness:** 3
**Presentation:** 3
**Significance:** 3
**Originality:** 3
**Overall Recommendation:** 5
**Confidence:** 4

**Summary:**

This paper investigates activation sparsity in large language models during inference and identifies representational instability as the key cause of performance degradation at high sparsity levels. The authors reframe the problem as a representation alignment task and propose a method called SPON to stabilize activation distributions without increasing inference overhead. Experiments conducted across various models and sparsity settings demonstrate that SPON effectively maintains model performance.

**Compliance With Llm Reviewing Policy:**

Affirmed.

**Final Justification:**

My concerns have been adequately addressed, so I am willing to recommend accepting this paper.

**Key Questions For Authors:**

1. The paper evaluates SPON across various sparsity ratios. However, it remains unclear if these theoretical ratios translate into tangible performance gains on actual hardware. Could the authors provide empirical measurements of end-to-end inference latency on standard GPUs to substantiate the claimed efficiency?

2. The authors emphasize that SPON is input-independent. Given that LLMs rely on dynamic responses to varied contexts, how does this static bias-based mechanism handle complex scenarios such as polysemous word disambiguation or long-range dependencies?

3. SPON is compared against LoRA and BitFit, which were not originally designed for recovering performance lost to sparsification. Would a more rigorous baseline be full-parameter fine-tuning of the sparse model on the same calibration data?

**Limitations:**

No. The authors have not adequately addressed the limitations of their research and its potential adverse societal impacts, and the suggested improvements are presented in the Key Questions section.

**Strengths And Weaknesses:**

Strengths：

1. The paper identifies representational instability as a key reason for the performance degradation of activation sparsity in LLMs. Reframing sparse activation as a representation alignment problem provides a clear conceptual perspective that helps explain the failure modes of prior approaches.
2. The proposed Spontaneous Neurons (SPON) mechanism is conceptually simple and easy to integrate into existing models. The injected vectors can be absorbed into bias terms after training, resulting in negligible inference overhead.
3. Experiments are conducted on several LLM backbones and across different sparsity levels, demonstrating that SPON consistently improves performance under high sparsity while maintaining computational benefits.

Weaknesses：

1. SPON requires a distribution-matching training stage with the dense model, which introduces extra training cost and may limit applicability in settings where the original dense model is unavailable.
2. This paper employs TEAL as the primary sparsification engine, it fails to explore the impact of diverse sparsification strategies on the effectiveness of SPON. This lack of cross-method evaluation limits the generalizability of the proposed approach.
3. Although the paper attributes the optimization objective of SPON to a Fisher-weighted approximation of the residual e(X), it fails to provide a clear connection between this mathematical approximation and 'knowledge preservation' at a semantic level.

---

> ### Author Rebuttal · Authors · 2026-03-26
>
> We sincerely thank reviewer b6pM for the thorough review and for recognizing our contributions. We hope the following responses adequately address the raised concerns.
>
> **W1:** We acknowledge this requirement, but note it is a standard assumption shared across virtually the entire efficient AI literature, including not only activation sparsity but also network pruning, quantization, and knowledge distillation, all of which require dense model access for calibration, scoring, or teacher supervision. Settings without dense models would equally preclude these studies. Moreover, SPON requires only forward access to the dense model and does not modify pretrained weights, unlike typical post-training approaches.
>
> **W2:** Since SPON operates purely in activation space and corrects the residual $e(X) = \mathbf{W}X - \mathbf{W}S(X)$ regardless of how $S(\cdot)$ is defined, it is by construction agnostic to the choice of sparsification engine. To validate this, we adapt WINA to Llama3-8B on language modeling task, and WINA-SPON can achieve **7.940** ppl better than 8.336 ppl from WINA.
>
> **W3:** The Fisher Information $H(z) = \text{diag}(p) - pp^\top$ assigns higher importance to output dimensions where the model's predictive distribution is most sensitive. Since our objective explicitly minimizes the KL divergence between dense and sparse outputs, it prioritizes matching this predictive behavior. By driving $\mathbf{W}\vec{\alpha}$ to approximate $e(X)$ along these high-Fisher directions, SPON preferentially compensates for sparsity-induced errors in the most semantically critical directions. Section 3.5 and Appendix G empirically corroborate this distributional drift mitigation in the latent space.
>
> **Q1:** Regarding inference latency, TEAL (Figure 3) and LaRoSA (Table 3) have already demonstrated the hardware efficiency of this sparsification paradigm, and our Tables 3 and 5 confirm that SPON introduces no additional FLOPs or MACs. To directly address the reviewer's question, we reproduce end-to-end decoding throughput on Llama3-8B with SPON across sparsity levels:
> |Sparsity|TPS|
> |-|-|
> |0.1|38.58|
> |0.2|41.45|
> |0.3|48.28|
> |0.4|48.07|
> |0.5|54.52|
> |0.6|66.67|
> |0.7|79.15|
> |0.8|90.29|
> |0.9|85.82|
>
> For reference, Llama3-8B without SPON achieves 56.60 tokens/sec at 50% sparsity under identical conditions, consistent with our Table 3. Minor numerical differences from Table 3 arise from prompt variation, but the conclusions remain unchanged: throughput scales with sparsity, and SPON introduces no measurable latency penalty, in agreement with TEAL and LaRoSA.
>
> **Q2:** Since $\vec{\alpha}$ is input-independent, it does not adapt to individual tokens. However, this is by design: SPON is not intended to replace input-dependent computation, but to restore the *global representational baseline* that activation sparsity erodes. The dynamic, context-sensitive computation remains entirely governed by the input-stimulated activations $S(X)$. The consistent improvements on tasks requiring polysemous disambiguation (CommonsenseQA) and long-range reasoning (BoolQ, MMLU) empirically confirm our claim. We also test Llama3-8B w/wo SPON on LongBenchV2 for long-context exploration. SPON can reach average **34.46%** slightly better than 32.99% from the non-SPON counterpart.
>
> **Q3:** Full-parameter fine-tuning is not a comparable baseline in our setting. First, it incurs training costs orders of magnitude higher than SPON, making the comparison fundamentally asymmetric. Second, fine-tuning on calibration data risks catastrophic forgetting of pretrained knowledge and overfitting to the calibration distribution, both of which undermine generalization. Third, even when allowed, it effectively re-optimizes the model rather than correcting sparsity-induced errors. Nonetheless, we compare against task-targeted post-training methods: LoRA and BitFit are included because they share structural similarity with SPON and isolate the contribution of activation-space learning. The key distinctions are:
>
> - **LoRA** introduces input-*dependent* low-rank corrections in weight space, which are less effective for correcting input-agnostic sparsity-induced residuals. As shown in Appendix A, activation sparsity is equivalent to dynamic neuron-level pruning, and Table 10 shows LoRA underperforms SPON.
> - **BitFit** is the closest structural analogue: in modern LLMs trained without bias terms, learning $\vec{b}$ with zero initialization is exactly BitFit. Section 4.1 and Section 2.4 show that learning via $\vec{\alpha}$, which constrains corrections to lie within the pretrained feature subspace spanned by $\mathbf{W}$, is superior, even with $16\times$ more bias parameters.
>
> Additionally, we compare against SCAP (lines 359–367), a statistics-based post-training activation pruning scheme targeting the same setting. SPON substantially outperforms SCAP at higher sparsity, and Table 2 provides further comparisons against current state-of-the-art methods.

---

> > ### Author Rebuttal · Reviewer_b6pM · 2026-04-03
> >
> > Very appreciate authors' effort. My concerns have been adequately addressed. I don't have any questions anymore. I have already adjusted my score.

---

> > > ### Author Response · Authors · 2026-04-03
> > >
> > > We are glad to see our rebuttal comments effectively resolve those concerns and are sincerely grateful for reviewer b6pM's appreciation on raising the score!

---

### Decision · Program_Chairs · 2026-04-30

**Decision:**

Accept (regular)

**Comment:**

This paper addresses LLM inference efficiency by studying activation sparsity. It identifies representational instability as a key reason for the performance degradation caused by activation sparsity and reframes it as a representation alignment problem. Finally, it proposes a technique named Spontaneous Neurons to address this problem. The experimental validation shows that the proposed technique achieves good performance under a high activation sparsity regime. Three reviewers declared that their concerns had been addressed during the rebuttal and recommended acceptance, while the only reviewer (ZWkh) who recommended weak reject did not engage in the discussion and did not acknowledge the rebuttal.